# Population genomics reveals an ancient origin of heartworms in canids
Rosemonde I. Power [1,24,25], Swaid Abdullah[2], Heather S. Walden[3], Guilherme G. Verocai[4], Tiana L. Sanders[4], Joe L. Luksovsky[4], Andrew R. Moorhead[5], Michael T. Dzimianski[6], Jeremy M. Foster [7], Michelle L. Michalski [8], Alicia Rojas [9,10], Samuel C. Chacón[11], Georgiana Deak[12], Andrei D. Mihalca[12], Patrizia Danesi[13], Elias Papadopoulos[14], Piyanan Taweethavonsawat[15,16], Dung Thi Bui [17,18], Anh Do Ngoc [19], Reuben S. K. Sharma[20], Simon Y. W. Ho [21], Stephen R. Doyle [22,26] ✉ & Jan Šlapeta [1,23,26] ✉

Heartworms (*Dirofilaria immitis*) are parasitic nematodes that cause significant cardiopulmonary-associated morbidity and mortality in canids worldwide. The global spread of heartworms is believed to have occurred alongside the dispersal of modern domesticated dogs over the past few hundred years. However, this conclusion has been based on limited sampling, both geographically and numerically. To test this theory more rigorously, we analyse the whole genomes of 127 adult heartworm specimens collected from mammalian carnivore hosts across Australia, the USA, Central America, Europe, and Asia. Here we show distinct genetic differences between heartworms from different continents, indicating a more ancient dispersal in canid hosts than previously recognised. Using genetic diversity and admixture analyses, we find an Asian origin for Australian heartworms, aligning with the arrival of dingoes into Australia via Asia thousands of years ago; however, we cannot exclude the alternate hypothesis that heartworms were also introduced from Asia in post-colonial times. Finally, the genetic relatedness between European and Central American heartworms suggests that modern dispersal, potentially associated with human colonisation of the Americas by Europeans, occurred with domesticated dogs. This work sheds light on the population dynamics and deep evolutionary history of a globally widespread parasite of veterinary significance.

The canine heartworm, *Dirofilaria immitis*, is a parasitic nematode that infects a broad range of mammalian carnivores (order Carnivora) across tropical, subtropical, and temperate regions worldwide[1]. Mosquitoes transmit larval stages of heartworms which, after infection, develop and reside in their host's heart and pulmonary arteries as adults[2]. Adult worms can cause lesions, vascular damage, pulmonary hypertension, and right-sided congestive heart failure, potentially leading to the death of their host[2]. This is of particular concern in domestic dogs, where the onset of disease can be prevented through the rigorous and lifelong application of parasiticides[3]. Such treatment regimes have led to the emergence of drug resistance[4,5] that, while currently only well described in the USA, threatens heartworm control globally if it were to spread. Heartworms are predicted to increase in prevalence and expand into new regions due to climate change, pet travel, and habitat alterations[6]. Therefore, defining the origins, distribution, and capacity for transmission will have important implications for implementing effective heartworm surveillance and control.

Current views on the evolutionary origins and subsequent radiation of heartworms worldwide are anthropocentric. Heartworms are believed to have originated in Europe[7] or Asia[2] and spread globally over the past 200–400 years via human-mediated dog movements. Accordingly, heartworms are presumed to form a single homogeneous global population. While some studies have supported this hypothesis by demonstrating low diversity and limited population structure amongst heartworms[8–11], these studies had limited sampling, both in terms of the numbers of sequences and geographic representation. The currently accepted view of a recent global dispersal also overlooks the deep co-evolutionary history that many parasites have shared with their hosts and the environments in which they reside. The broad host range of heartworm potentially indicates a long co-evolutionary history with mammalian carnivores, especially canids[12,13], a group that dates back ~40 Mya[14], predating humans (~2.8 Mya)[15] and the domestication of modern dogs (14–40 kya)[16–18]. Given also that mosquito vectors, which are essential for heartworm transmission, have existed since

the mid-Cretaceous (~106 Mya)[19], it is possible that heartworms radiated long before human involvement. We propose that heartworms offer a compelling case for assessing the degree to which anthropogenic activities have influenced parasite distribution.

Analyses of genomic data have significant potential to identify the historical origins of heartworms and infer their global dispersal routes, as demonstrated for other helminths[20–22]. Sequencing adult heartworms is recommended for obtaining high-quality data[23]; however, obtaining adults from the heart requires invasive surgery or post-mortem, which are often not feasible. Such sampling is especially difficult in parts of Asia and Africa, where cultural perceptions of dogs as unclean or carriers of disease can restrict sampling opportunities. In light of these challenges, here we present the most extensive population genomics study of heartworms to date. Using whole genome sequencing of 127 adult worms obtained from nine countries on four continents, we analyse the diversity, genetic structure, and population demographics of modern heartworms to infer their historical origins and dispersal.

Our population genomic analyses of heartworm parasites reveal a global diversity that has been shaped by both ancient and recent biogeographic events. We observed strong continental population structure, likely influenced by the distribution of ancestral canid hosts throughout time. We detected genetic admixture between Asian and Australian heartworms, suggestive of translocation of worms with the arrival of dingoes in Australia or potentially, through modern movement of dogs from Asia. We also provide evidence of recent human-driven dispersal, potentially linked to the movement of domesticated dogs from Europe to Central America during colonisation.

## Results and discussion
### Genomic diversity of heartworms from multiple continents, countries, and hosts

We analysed whole-genome sequencing data from 127 adult heartworms (*D. immitis*) sampled from nine countries in five regions: Australia, North America (USA), Central America (Costa Rica, Panama), Europe (Greece, Italy, Romania), and Asia (Malaysia, Thailand) (Fig. 1a). Samples were primarily collected from domestic dogs (*n* = 115), but we also sampled cats (*n* = 4), foxes (*n* = 4), a ferret (*n* = 1), a golden jackal (*n* = 1), a leopard (*n* = 1), and a wildcat (*n* = 1). Across the full cohort, inclusive of outgroup samples, we obtained an average coverage per sample of 72.72× for the nuclear genome, 12,788.18× for the mitochondrial genome, and 1014.25× for the *Wolbachia* endosymbiont genome (Supplementary Data 1, Supplementary Figs. 1–4). Variant calling, joint genotyping, and filtering of the full cohort revealed 301,310 single nucleotide polymorphisms (SNPs) and 80,514 indels in the nuclear genome, along with 498 SNPs and 24 indels in the mitochondrial genome, and 25,473 SNPs and 1381 indels in the *Wolbachia* genome; within *D. immitis* samples, 301,004 nuclear SNPs (3.4 SNPs per kb), 57 mitochondrial SNPs (4.1 SNPs per kb), and 360 *Wolbachia* SNPs (0.4 SNPs per kb) were found. The SNP frequencies in the nuclear and mitochondrial genomes are comparable, whereas the frequency in the *Wolbachia* genome is considerably lower. These frequency data are comparable to those of other *Wolbachia*-infected filarial worms, such as *Onchocerca volvulus*[24] and *Wuchereria bancrofti*[22].

### Nuclear variants reveal distinct continental partitioning of heartworms

Analyses of the heartworm cohort's genetic diversity revealed distinct groups of samples based on broad geographical regions. Principal component analysis (PCA) using 218,158 high-quality autosomal SNPs revealed four clusters of samples corresponding to the continents from which the samples were collected (Fig. 1b and Supplementary Fig. 5; PC1 variance: 13.83%, PC2 variance: 10.68%). The third and fourth PCs showed a similar pattern of geographical clustering (Supplementary Fig. 6a; PC3 variance: 6.16%, PC4 variance: 4.42%); however, although the signal was comparatively weaker, the European and Central American samples formed separate clusters. Similar findings were observed in a previous study of 31 heartworm

samples from Australia, Italy, and the USA, where samples from each country formed distinct clusters[25]. Independent PCAs of SNPs from each chromosome found the same continental clustering pattern across all chromosomes, including the sex-linked X chromosome (Supplementary Fig. 6b–f), suggesting that this is a genome-wide rather than region-specific observation. The distinct geographical partitioning in our nuclear data was not supported by the mitochondrial (Supplementary Fig. 7a) or *Wolbachia* data (Supplementary Fig. 7b), likely due to the limited number of variant sites within each dataset. This contrast between the strong nuclear and weak mitochondrial population structure was particularly surprising, however, the underlying biological cause remains unknown.

Analyses of genome-wide nucleotide diversity (Pi) further demonstrated differences between heartworm populations across continents. Pi differed significantly between all populations (Wilcoxon Rank Sum test, $P < 0.05$) (Fig. 1c). The highest Pi was observed in the USA (median window-averaged Pi $= 7.77 \times 10^{-4}$), while the lowest was in Asia ($3.12 \times 10^{-4}$). The genetic diversity of our USA samples aligned closely with that of a previous *D. immitis* study (median window-averaged Pi $= 7.36 \times 10^{-4}$)[25]. However, our Australian samples (median window-averaged Pi $= 6.29 \times 10^{-4}$) exhibited higher genetic diversity than those reported previously ($3.9 \times 10^{-5}$)[25], likely due to the larger and geographically broader sampling here. Overall, the nucleotide diversity of our *D. immitis* samples is comparable to that of the filarial worm *W. bancrofti* (Pi[mean] $= 2.4 \times 10^{-4}$)[22] but lower than that of *O. volvulus* (Pi[mean] $= 4.0 \times 10^{-3}$)[24]. Each *D. immitis* population exhibited different rates of linkage disequilibrium (LD) decay and baseline LD (Fig. 1d), with Asia and the USA showing the highest and lowest LD, respectively. Estimates of admixture provided further support for continental partitioning in heartworms. To infer the ancestry proportions of samples, we used NGSadmix with a range of *K* clusters (*K* = 2–10; Supplementary Fig. 8). At *K* = 7, Australian samples showed the most diverse admixture patterns with strong evidence of shared ancestry with Asia, whereas USA, Europe, and Central America were largely distinct with some evidence of shared ancestry in a minority of samples (Fig. 1e). The strong population structure in the nuclear genome of heartworms suggests that spatial spread has been restricted during the species' history, resulting in distinct genetic profiles on each continent.

### Inference of transmission between diverse heartworm hosts

Heartworms are obligate, vector-borne parasites whose successful establishment in a new environment depends on three key requirements. The first requirement is the presence of a suitable definitive host in which heartworms can develop, establish in the pulmonary arteries and the heart, and sexually reproduce to generate microfilariae, which circulate in the host's bloodstream. The second requirement is a suitable mosquito vector that can ingest the microfilariae and support their development until they reach their infective larval stages; at least 17 mosquito species, including members of *Aedes*, *Anopheles* and *Culex* spp., are confirmed vectors in Europe[26], and perhaps as many as 60 different mosquito species in total[27], have been identified as competent vectors of *D. immitis*. The final requirement relates to the conditions needed for larval development within the mosquito, which is both temperature- and time-dependent. The rate of parasite development within the mosquito, from ingested microfilaria to infective stage larvae, is positively associated with outside ambient temperature, with a minimum temperature of 14 °C[28,29]. As such, warmer climates, including those induced by climate change, would accelerate this developmental stage inside the mosquito vector. This complex interplay between host, vector, and climate has enabled heartworms to infect both domestic and wild animals on almost every continent of the world.

Although most heartworm samples in this study were obtained from dogs, the analysis of heartworm genomes from foxes, cats, a golden jackal, a ferret, and a leopard allowed us to evaluate genetic variation among different host species. We found that geography rather than the host species explained variation in the global heartworm population, evidenced by samples from cat and ferret nested within samples from dogs in the USA, fox

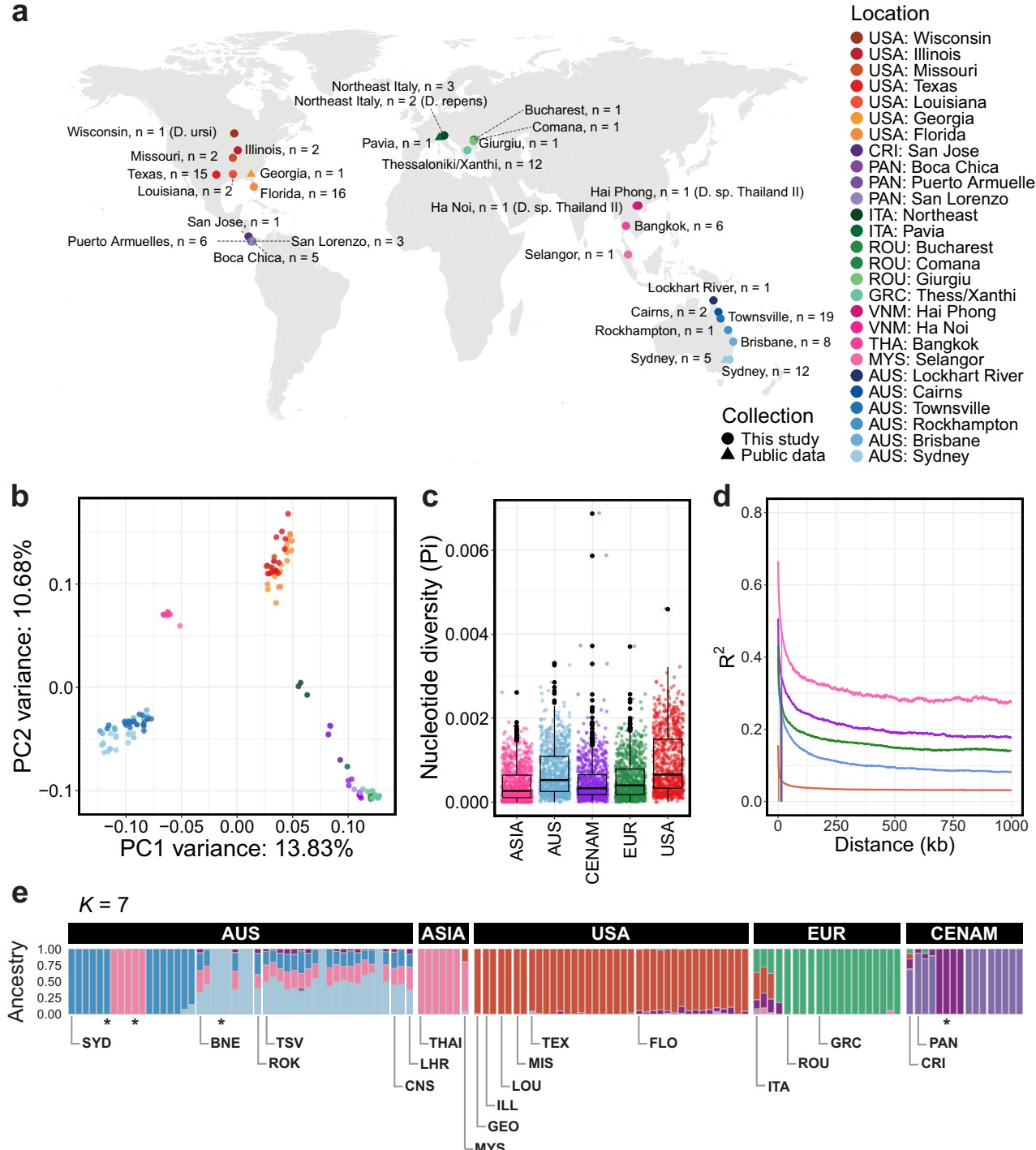

**Fig. 1 | Global sampling and population structure of heartworms. a** World map showing the location of adult heartworm specimens used in the study. Adult *Dirofilaria ursi*, *D. repens*, and *D.* sp. 'Thailand II' were included as outgroup samples. **b** Principal component analysis (PCA) of 218,158 autosomal single nucleotide polymorphisms (SNPs) from 124 heartworms. **c** Box plots showing the nucleotide diversity (Pi) distribution within broad geographical regions, with each point representing a 100 kb sliding window. **d** Linkage disequilibrium (LD) decay plot for the heartworm cohort, grouped by broad geographic region. Vertical lines show the distance between SNPs where the LD is half the maximum value. **e** Admixture

inferred using NGSadmix with seven clusters (K). Each vertical bar represents an individual sample, with colours representing the clusters. Labels at the top of the bar charts indicate the broad geographic origins of samples, while those at the bottom show their more specific locations. An asterisk (*) denotes a replicate of the preceding sample. USA United States of America (Georgia: GEO, Illinois: ILL, Louisiana: LOU, Missouri: MIS, Texas: TEX, Florida: FLO), CRI Costa Rica, PAN Panama, ITA Italy, ROU Romania, GRC Greece, VNM Vietnam, THAI Thailand, MYS Malaysia, AUS Australia (Sydney: SYD, Brisbane: BNE, Rockhampton: ROK, Townsville: TSV, Cairns: CNS, Lockhart River: LHR).

and dogs in both Australian and Italian groups, and dogs and jackal, wildcat and leopard from Europe (Fig. 2a). Analyses of nucleotide diversity (Pi), absolute nucleotide divergence ($D_{XY}$), and genetic differentiation ($F_{ST}$) provided additional insights on the variation within and between different

host species. The heartworm sampled from the ferret had the highest diversity (median window-averaged Pi = $1.27 \times 10^{-3}$) (Fig. 2b) and the greatest divergence from other hosts (Supplementary Fig. 9a). In contrast, the samples from the golden jackal, leopard, and wildcat had the lowest

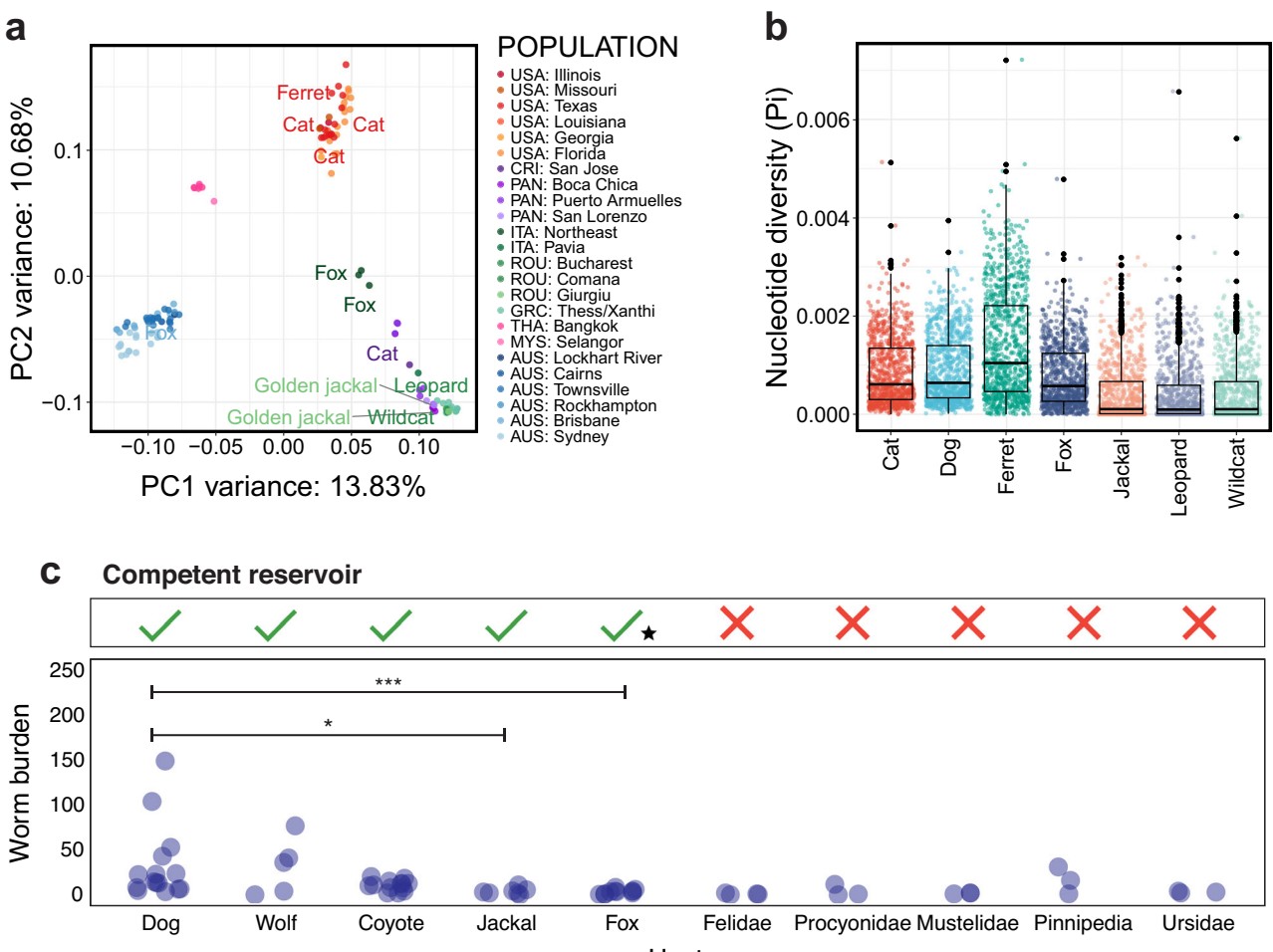

**Fig. 2 | Variation in the genetics and competency between heartworm hosts.**
**a** Principal component analysis (PCA) of heartworm nuclear diversity from Fig. 1b, with non-dog hosts labelled. The geographic origin of samples is indicated on the right (i.e. COUNTRY: city or region). Country abbreviations: USA = United States of America; CRI = Costa Rica; PAN = Panama; ITA = Italy; ROU = Romania; GRC = Greece; THA = Thailand; MYS = Malaysia; AUS = Australia. **b** Box plots show the distribution of nucleotide diversity (Pi) within host populations. Each data point represents a sliding window size of 100 kb. **c** Support for the heartworm life cycle and adult worm burdens in carnivoran hosts. The top panel highlights hosts that are frequently microfilaremic and are hence considered competent reservoirs for heartworms. A black star indicates cases where microfilaremia has been reported in the host, but they are generally considered to pose a low risk of being competent reservoirs. The bottom panel is a plot showing adult heartworm burdens across various hosts. Each point represents data from a study, sourced from the literature (Supplementary Data 2). Significant differences in worm burdens among competent reservoirs were assessed using a Kruskal–Wallis test with Dunn's multiple comparisons. $P$-values ≤ 0.05 were considered statistically significant ($^*$≤0.05, $^{**}$≤0.01, $^{***}$≤0.001).

diversity (median window-averaged Pi = $1.46–2.09 \times 10^{-4}$) and were the least divergent from each other compared with all other host pairs. However, only a single heartworm was sampled from each of these hosts, and the variation observed in the jackal, leopard, and wildcat samples may be confounded as they all originated from Romania. $F_{ST}$ values between host pairs were generally low, with foxes and cats showing the highest genetic differentiation from the jackal, leopard, and wildcat (Supplementary Fig. 9b). Collectively, these data suggest an absence of obvious host-specificity and that transmission is likely between domestic animals and wildlife. However, further geographic sampling from non-dog hosts is needed to exclude the possibility of host-specific adaptation.

We next questioned which host may have harboured and dispersed heartworms before the emergence of modern dogs. We hypothesised that longer co-evolution between parasites and hosts would result in the latter being able to carry higher parasite loads. Felids (e.g. domestic cats, wild cats, snow leopards), procyonids (e.g. raccoon dogs), mustelids (e.g. European badger, ferrets), pinnipeds (e.g. harbour seal, brown fur seal), and ursids (e.g. brown bear, black bear) have been described as being infected with heartworms

(Supplementary Data 2); however, they typically carry fewer adult worms and fail to support the parasite's entire life cycle, and so they are not considered primary reservoirs[30–33] (Fig. 2c). In contrast, wild canids are commonly microfilaremic and have high worm burdens. Wolves and coyotes have worm burdens similar to dogs, while jackals and foxes have significantly lower burdens than dogs (Kruskal–Wallis test with Dunn's multiple comparisons; Dog vs Jackal: $P \leq 0.05$, Dog vs Fox: $P \leq 0.001$) (Fig. 2c). These findings imply that canids, particularly ancestral wolves or coyotes, were most likely the primary hosts for heartworm during ancient times[34].

## Coevolution of canids and heartworms: distinguishing modern from ancient origins and dispersal
Previous work had proposed that canids originated in North America ~40 Mya when the continent was completely isolated[14], whereas mosquitoes originated earlier in the mid-Cretaceous (~106 Mya)[19]. Phylogenetic analyses of mitochondrial DNA suggest that heartworms diverged from their closest relatives in the genus *Onchocerca* ~25 Mya during the Oligocene[35], i.e. when canids and mosquito vectors were already well-established.

Together, this timeline supports a long co-evolutionary history between heartworms and ancient canids.

The formation of the Bering Land Bridge and the Isthmus of Panama during the Miocene and Pliocene facilitated the first migrations of canids into Eurasia (~7 Mya)[14] and South America (3.5–3.9 Mya)[36], respectively. Throughout history, fluctuations in sea level have repeatedly exposed and submerged the Bering Land Bridge. Fossil records and genomic data provide robust evidence that these periods of exposure enabled ancient canids to migrate between Asia and North America, facilitating population connectivity and supporting their widespread presence across the Northern Hemisphere. Examples include Pleistocene wolves, which were highly connected across North America and Eurasia[16,37,38], *Xenocyon lycaonoides*, a large canid distributed across North America and Eurasia in the mid-Pleistocene[39] and *Eucyon*, a jackal-sized canid distributed across North America, Eurasia, and Africa in the late Miocene[40]. The widespread presence of ancient canids across continents, together with evidence of their population connectivity, suggests that heartworms may have similarly formed highly connected populations alongside their hosts at that time.

The current hypothesis for the distribution of heartworms around the world is that it has been largely influenced by human movement in contemporary times. Alternative hypotheses suggest that heartworms spread with humans during or shortly after dog domestication (~14–40 kya), or that they spread with wild canids before domestication. To further explore the demographic history of heartworms, we inferred their effective population (Ne) size histories using SMC++ over the last 1 million years. Given the domestication hypothesis, we broadly characterise the population demographics into three periods: post-domestication (<14 kya), domestication (14–40 kya), and pre-domestication (>40 kya). Contemporary estimates of Ne after domestication (Fig. 3a) were consistent with current levels of genetic diversity (Fig. 1c) in each population, with USA and Australian populations showing the highest Ne and Pi, and Asia the lowest. Differences between populations in demographic analyses broadly support the $D_{XY}$ and $F_{ST}$ results (Supplementary Fig. 10), which indicate the greatest differentiation and, therefore, infer population splits from Asia, which diverged from the populations of Europe, the USA, and Central America. Considering the variation in population demographic curves >40 kya (Fig. 3a), we propose that there were substantial differences in some populations that were evolving independently before dog domestication[16–18]. Given that modern global human movement would drive populations to be more similar, the exacerbation of differences in Ne post-domestication would further suggest that these differences in populations existed before domestication.

If heartworms were once one population, when did they diverge, and why? Our estimates of Ne do not extend earlier than 1 million years ago. However, between 1 million and ~50,000 years ago, significant climatic events correlate with changes in the heartworm population demographics. For example, we observe a substantial increase in Ne of all populations roughly aligning with the last interglacial period (Fig. 3a), when Earth experienced a climate as warm or even warmer than today (116–130 kya)[41]. Such an environment could have been suitable for the development and transmission of heartworms via mosquito vectors. Following the warm climates of the last interglacial period, the Earth began to cool. By Marine Isotope Stage 4 (58–72 kya), large continental ice sheets had formed across North America, Beringia, and northern Eurasia[42]. These ice sheets separated ancient canid populations[37,43], potentially fragmenting the associated ancient heartworm population. This time period was associated with a fall in Ne, particularly in the USA samples, but for all populations, a considerable decrease was observed immediately before or during dog domestication, which occurred shortly afterwards (Fig. 3a–d). This fragmentation would have reduced gene flow for extended periods, explaining the continental grouping of samples in our study. Further insights might be obtained by exploring the presence of heartworm DNA in the remains of ancient canids; if successfully extracted, these ancient worm DNA sequences may fill in the temporal and geographic gaps needed to understand heartworm divergence

in more detail. Similarly, further sampling in unstudied regions, including South America and Africa, and finer sampling within Asia, would enhance our understanding of heartworms' evolutionary history.

Considering global warming and cooling cycles, the fragmentation of canid and heartworm populations would have occurred multiple times. We found evidence of secondary admixture between heartworms from Asia and the USA after the initial population divergence, evident from both the admixture analyses and gene flow between populations (Fig. 4a; delta $m = 4.1$; all tree migration edges are presented in Supplementary Fig. 11). $D$-statistics (Fig. 4b, c) provided further support for excessive allele sharing between the USA and Asia relative to that between the USA and other regions (Fig. 4b). This admixture may reflect multiple waves of dispersal with canids across the Bering Land Bridge in the Late Pleistocene during periods of ice sheet contraction[16] between 11–30 kya and 60–70 kya. Our population demographic data provide some support for this hypothesis, at least during 11–30 kya, when USA and Asian populations experienced a close overlap in Ne before undergoing rapid and distinct divergence thereafter (Fig. 3b).

## The origin and spread of heartworms in Australia

Heartworms are endemic in Australia, but the mechanisms and timing of their arrival in the country have remained an enigma. Consistent with the modern human dispersal hypothesis, one means of introduction could have been after European colonisation, which began ~235 years ago. However, considering our genetic evidence of older, canid-mediated dispersal, another likely scenario is that heartworms were brought to Australia via dingos. The dingo, an ancient lineage from Asia, was the only canid present on the Australian continent until European settlement. Recent morphometric analysis indicates an East Asian origin of dingoes via Melanesia[44], whereas genetic studies suggest that dingoes likely came from Island Southeast Asia and were brought to Australia by seafarers ~4 kya[45].

Multiple lines of evidence support the introduction of heartworms into Australia from Asia. Both admixture and TreeMix analyses highlighted the close relationship, evidenced by partial or complete admixture profiles (Fig. 1e) and mutual monophyly (Fig. 4a) between Asian samples and samples collected along the East Coast of Australia (Fig. 5a). Calculations of $D_{XY}$ showed that AUS-ASIA (median window-averaged $D_{XY} = 5.81 \times 10^{-4}$) was the least divergent pair, suggesting that Australian worms are more closely related to those in Asia than elsewhere (Supplementary Fig. 10a). $D$-statistics solidified this relationship by showing excessive allele sharing between the two continents relative to that between Australia and other regions (Fig. 5b). Finally, samples from Australia and Asia exhibit very similar population demographic trajectories of Ne until ~10 kya, which begin to diverge in the post-domestication period, consistent with the timing of the arrival of dingoes. Together, our findings align with an Asian origin of Australian heartworms, possibly transported with dingo hosts (Fig. 3c). We note that our sampling of Asian heartworms was from Thailand and Peninsular Malaysia, which are geographically distinct from Indonesia, from which dingoes would have entered Australia from ~4 kya, potentially explaining the small discrepancy in the estimated time of divergence. Sampling across a broader geographic region, particularly throughout the Indonesian archipelago and Melanesia, could bridge the gap between our Australian and Asian samples and provide further granularity of the Asia–Australia divergence, potentially coinciding more closely with the timeline of dingo arrival. An alternative, plausible scenario is the post-colonial introduction of heartworms into Australia through the importation of modern dog breeds via Asia. Importation of infected dogs from regions such as China or Japan could also explain the genetic divergence we currently observe between our Australian and Asian (represented only by Thailand and Malaysia) samples. If the genetic link between Asia and Australia is due to modern translocation, we might expect genetic links to other regions, not just Asia, which we do not find. Nonetheless, broader sampling across the Asian continent is needed to determine the relative genetic contribution of heartworms from ancient dingoes to more recent importation of modern dog breeds from Asia.

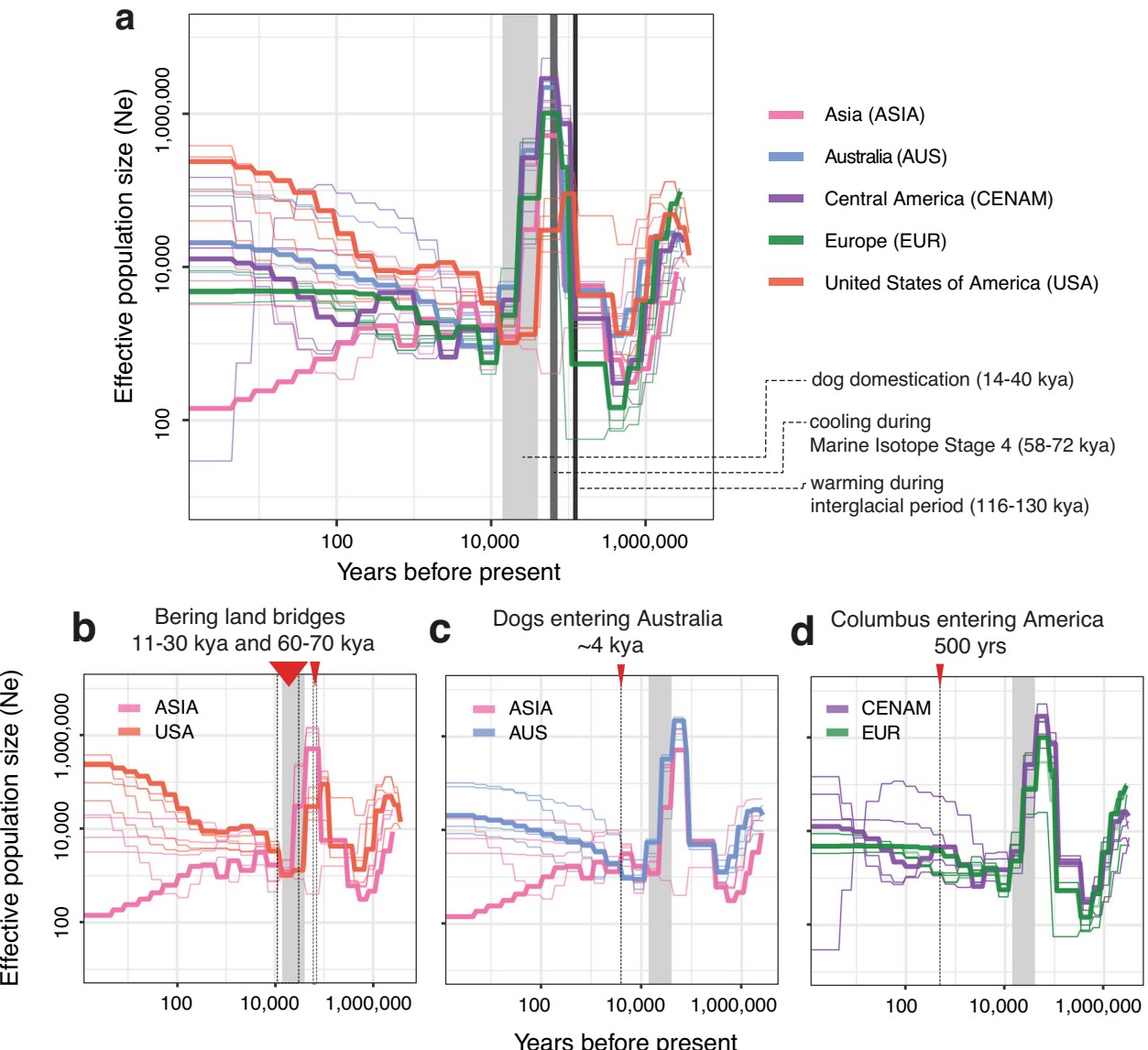

**Fig. 3 | Demographic history and host competency for heartworms. a** Effective population size history of heartworm populations inferred using SMC++ based on a 2.5-year generation time. Each thick line represents the effective population size of heartworms on each continent over time. Thin lines represent jackknife sampling of chromosomes, iteratively removing one of four autosomal chromosomes, to visualise variance in Ne. The light grey box spans ~14–40 kya, highlighting the previously suggested period of dog domestication, as well as dark grey boxes highlighting global cooling (58–72 kya) and warming (116–130 kya) periods. **b–d** Reconfigured datasets from **a**, highlighting key pairwise comparisons to explore hypotheses presented in the text. **b** Comparison of the effective population size histories of heartworm populations from Asia and the USA. The red arrows and dotted lines represent periods of the Bering Land Bridge exposure. **c** Comparison of effective population size histories of Asian and Australian heartworm populations. The red arrow and dotted lines indicate the suggested period of dingo introduction in Australia. **d** Comparison of effective population size histories of Central American and European heartworm populations. The red arrow with a dotted line indicates the period of European colonisation of the Americas.

Considering the extensive sampling along the north–south transect of eastern Australia, we sought to identify factors that might explain the current genetic diversity. Australian heartworms were the second most genetically diverse population globally (Fig. 1c). Within Australia, diversity was highest in Lockhart River (median window-averaged Pi = 8.87 × 10⁻⁴) and lowest in Rockhampton (2.60 × 10⁻⁴) (Fig. 5c). Heartworms from the northern state of Queensland were highly admixed, with two unique sources of ancestry that were largely absent from other continents (Fig. 1e). Interestingly, ten of 16 heartworms from Sydney, the southernmost site, had a single fixed ancestry that was detected in Queensland samples. Another four Sydney heartworms had a different fixed ancestry identical to that found in Asia, potentially reflecting modern movement, though the exact cause remains unclear.

Previously, it was believed that heartworm cases in temperate Sydney and broader New South Wales were primarily caused by infected pets or mosquitoes travelling down from tropical and subtropical Queensland[46], where climates are more favourable to heartworm transmission. However, our nuclear PCA showed a large cluster of samples primarily from Queensland, alongside two distinct clusters from Sydney (Fig. 5d). This suggests that Sydney heartworms may not necessarily originate from Queensland, as previously believed. Further investigations are required to determine whether there are local endemic populations of heartworms in Sydney. Differences in $D_{XY}$ and $F_{ST}$ between Australian heartworm subpopulations were subtle, showing no clear relationship between genetic and geographical distance as might be expected for a vector-borne parasite (Fig. 5e, f).

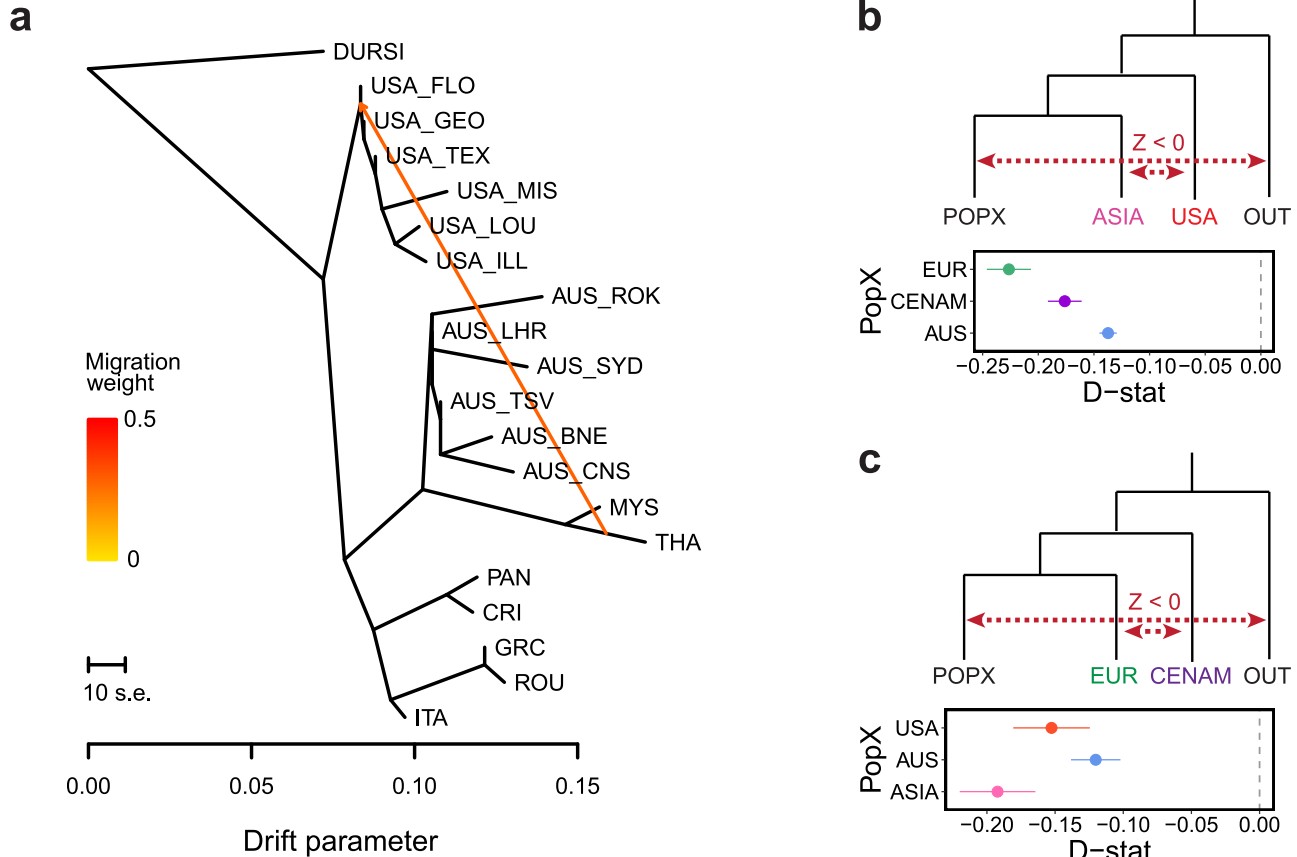

**Fig. 4 | Transcontinental admixture in heartworms. a** Maximum-likelihood tree of heartworm generated using TreeMix, with *Dirofilaria ursi* (DURSI) as an outgroup and one migration edge. **b, c** *D*-statistics for hypothetical allele-sharing scenarios between heartworm populations using Admixtools qpDstat with 20,392 single nucleotide polymorphisms (SNPs) and *D. ursi* as the outgroup (OUT). *D*-statistics test for admixture between four populations (W, X, Y, Z). A Z-score (*D*-statistic / standard error) is calculated, which provides information about the direction of gene flow (red dotted lines) (Z-score > 0 = gene flow between either [W and Y], or [X and Z]; Z-score < 0 = gene flow between either [W and Z] or [X and Y]). For all data presented, the Z score was significant (|Z| > 3). Error bars show the standard error.

## Modern heartworm dispersal between Europe and Central America

Our genomic analysis revealed a close genetic relationship between worms in Europe and Central America. This finding was most evident in our PCA of nuclear SNPs, where samples from Greece, Italy, and Romania clustered with those from Costa Rica and Panama (Fig. 1b). In addition, estimates of $D_{XY}$ revealed that EUR-CENAM (median window-averaged $D_{XY} = 5.93 \times 10^{-4}$) was the second least divergent compared with other populations. Admixture analysis using NGSadmix further supported this trend as the European and Central American samples shared high levels of ancestry at lower *K* values (*K* = 2–4; Supplementary Fig. 8). Although increasing *K* helped to differentiate worms from these two continents, low levels of admixture were still present in some samples (Fig. 1e). These findings were corroborated using *D*-statistics which provided strong evidence of excessive allele sharing between heartworms from Central America and Europe relative to that between Central America and other regions (Fig. 4c). Finally, Europe and Central America shared considerable overlap in Ne over deep evolutionary timescales (Fig. 3d). These findings support a modern migration event between European and Central American heartworms which took place after dog domestication.

One plausible modern migration event between Europe and Central America was the transatlantic transportation of domestic animals, including dogs, during the European colonisation of the Americas, which began in the late 15th century. Animals introduced to the Americas during the Age of Discovery were primarily livestock species, including cattle, sheep, pigs, goats, horses, and mules. However, dogs such as greyhounds and mastiffs were also introduced for hunting, guarding, protection, shepherding, and battle purposes[47–49]. The introduction of European dogs into the Americas is perhaps best documented from Columbus' voyages, with historical records indicating the presence of at least 20 dogs on board during the second voyage to Hispaniola[47]. Therefore, infected dogs from Europe may have transported local strains of this parasite to the Americas. Alternatively, the shared ancestry between European and Central American heartworms could result from the more recent migration of humans and their pets.

## Conclusions

Our transcontinental genome-wide analysis of heartworm parasites has revealed a much deeper origin and dispersal history than previously understood, allowing us to propose new hypotheses on the evolutionary history of heartworms (Fig. 6). Rather than being solely driven by human-mediated dispersal of dogs over the past few hundred years, we found that ancestral canid hosts played a pivotal role in the evolution and dissemination of heartworms around the world. Heartworms have evolved to exploit a unique niche in the heart and bloodstream of canids, an environment that has likely remained unchanged over millions of years between hosts. In the absence of niche competition, heartworms could be a relic of the past, which, until recently, had little need for further adaptation.

Anthropogenic influence is, however, likely to shape heartworm diversity more rapidly than ever before. For example, extensive drug treatment is already associated with resistance, particularly in the USA; climate change is expected to shift populations of parasites and their vectors into higher latitudes; and increasing global connectivity of people and their pets will all influence the distribution and adaptation of heartworms. Therefore, understanding the global genomic landscape of these parasites, as

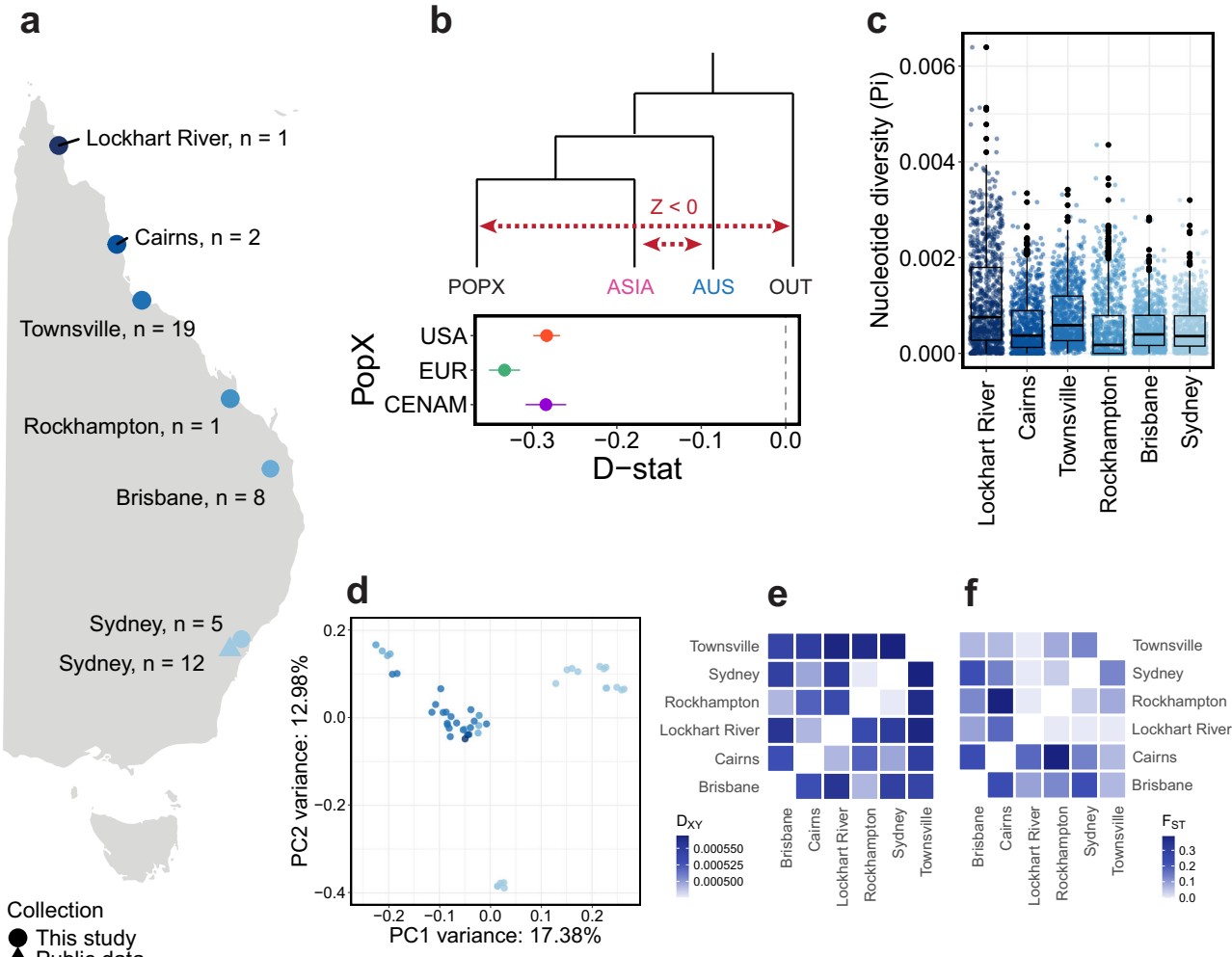

**Fig. 5 | Gene flow and population structure of Australian heartworms.**
**a** Distribution of adult heartworm samples ($n = 48$) collected along the east coast of Australia. **b** $D$-statistics showing excessive allele sharing between Australia (AUS) and ASIA relative to that between AUS and other regions (POPX). Admixtools qpDstat was used to analyse 20,392 single nucleotide polymorphisms (SNPs), with *Dirofilaria ursi* as the outgroup (OUT). $D$-statistics test for admixture between four populations (W, X, Y, Z). A Z-score ($D$-statistic / standard error) is calculated, which provides information about the direction of gene flow (red dotted lines) (Z-score > 0

= gene flow between [W and Y], or [X and Z]; Z-score < 0 = gene flow between [W and Z] or [X and Y]). For all data presented, the Z score was significant ($|Z| > 3$). Error bars show the standard error. **c** Box plots showing the nucleotide diversity (Pi) distribution from dog hosts per city. Each data point represents a 100 kb sliding window. **d** Principal component analysis (PCA) of 131,508 nuclear SNPs in the Australian cohort. **e** Absolute nucleotide divergence ($D_{XY}$) and **f** genetic differentiation ($F_{ST}$) between Australian cities using a median of 100 kb sliding windows. Only heartworms from dog hosts were included in the analyses for **c**, **e**, **f**.

demonstrated here, is vital for developing effective surveillance and control strategies in the future. Our data indicate that surveillance and control strategies should account for geographical variation between parasite populations and consider transmission within and between populations to maximise sustainable parasite control.

## Methods
### Sample collection
A total of 120 individual adult *D. immitis* were obtained from Australia ($n = 43$), the USA ($n = 37$), Panama ($n = 14$), Greece ($n = 12$), Thailand ($n = 6$), Italy ($n = 3$), Romania ($n = 3$), Costa Rica ($n = 1$), and Malaysia ($n = 1$). The sampling regime aimed to collect adult *D. immitis* from a wide geographical range. Most of these samples were obtained from dogs (*Canis lupus familiaris*; $n = 108$) but also included some from cats (*Felis catus*; $n = 4$), foxes (*Vulpes vulpes*; $n = 4$), a ferret (*Mustela putorius furo*; $n = 1$), a golden jackal (*Canis aureus*; $n = 1$), a leopard (*Panthera pardus*; $n = 1$), and a wildcat (*Felis silvestris*; $n = 1$) (metadata for all samples is presented in Supplementary Data 3). To provide outgroup samples for our analyses, we collected adult *D. repens* ($n = 2$) from dogs in Italy, *Dirofilaria* sp. 'Thailand

II' ($n = 2$) from humans (*Homo sapiens*) in Vietnam and *D. ursi* ($n = 1$) from an American black bear (*Ursus americanus*) in the USA (Supplementary Data 1).

All parasites used in this study were opportunistically collected and preserved by veterinarians or researchers during surgery or extracted from deceased dogs during postmortem, which were then subsequently donated to this study. We have complied with all relevant ethical regulations for animal use.

### DNA extraction and whole genome sequencing
*Dirofilaria* specimens were frozen at $-20\,°C$ or stored in ethanol upon collection and sent to the University of Sydney, Australia. A 1–2 cm segment between the first and second third of the anterior portion of the worms was excised using a sterile scalpel blade and placed in a 1.5 mL Eppendorf tube.

For ethanol-preserved samples, worm segments were centrifuged at maximum speed for 30 s using a benchtop centrifuge (Eppendorf, Australia) and excess ethanol was removed. The segments were washed in 500 mL of phosphate-buffered saline (PBS) and placed on a heat block (Heat Block Eppendorf Thermomixer Comfort or Heat Block Major Science Dry Bath

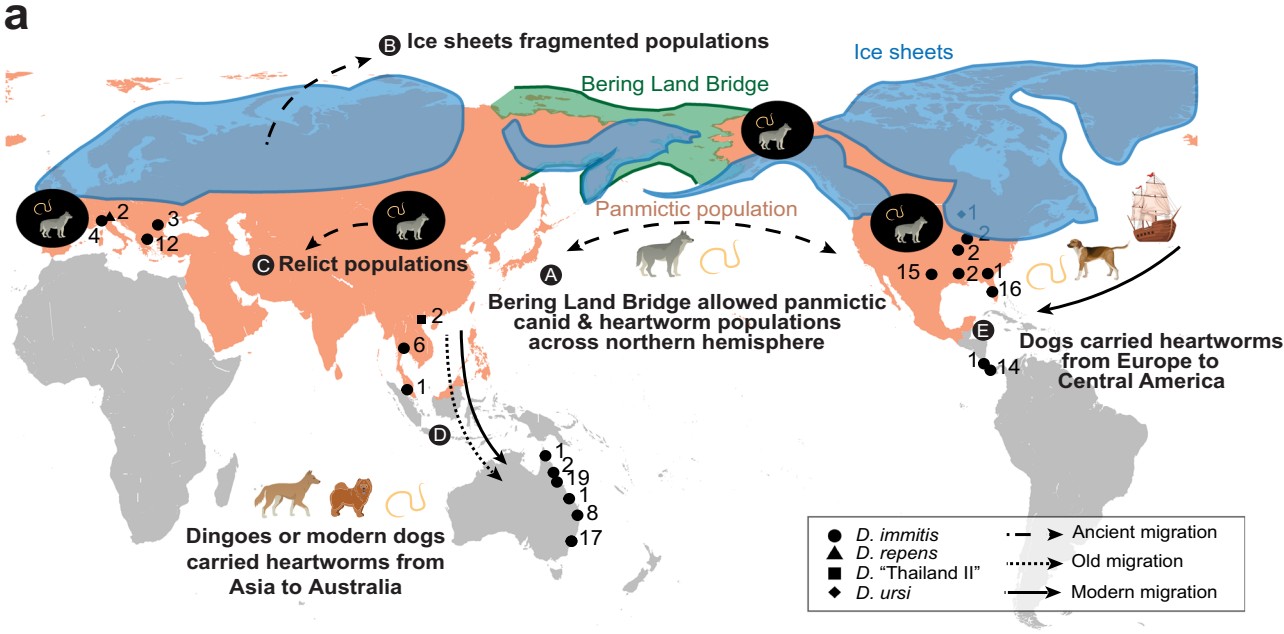

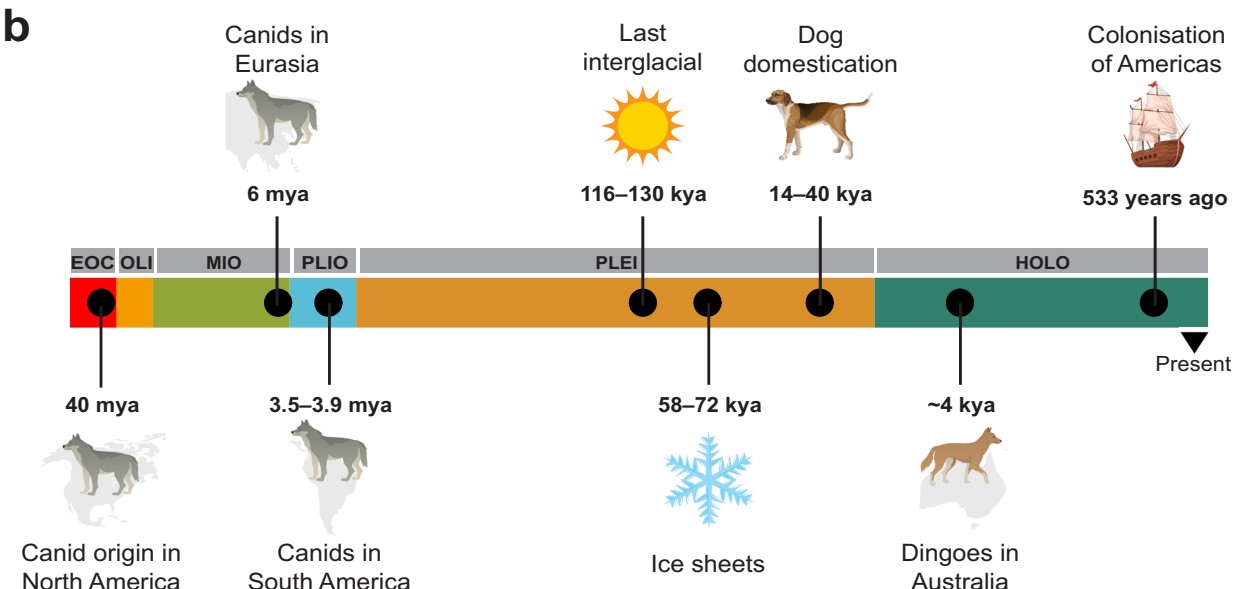

**Fig. 6 | A new evolutionary paradigm of heartworms throughout history.**
**a** Hypothetical heartworm dispersal scenarios supported in the current study. Pink regions represent a panmictic population of ancient canids harbouring heartworms across North America and Eurasia. The Bering Land Bridge is shaded in green, and continental ice sheets are shown in blue. Events are in chronological order (A–E), with arrows illustrating ancient (dashed), old (dotted), and modern (solid) migration events. Small black shapes with adjacent numbers indicate the geographic origin of samples collected in this study and their sample size. The legend specifies the parasite species, including the heartworm *Dirofilaria immitis* (*n* = 127) and outgroups *D. repens* (*n* = 2), *D.* sp. 'Thailand II' (*n* = 2), and *D. ursi* (*n* = 1). **b** Timeline of key canid evolution and climatic events. Horizontal bars represent Epochs: HOLO = Holocene; PLEI = Pleistocene; PLIO = Pliocene; MIO = Miocene; OLI = Oligocene; EOC = Eocene. The time scale was log-transformed to enhance the visualisation of events. Images used in this figure were obtained from Adobe Stock and are used under license.

Incubator) at 60 °C for 20–40 min, depending on their thickness, to evaporate any remaining ethanol.

For frozen samples, worm segments were defrosted and washed three times in PBS. Genomic DNA was extracted from the tissue segments using a Monarch® Genomic DNA Purification Kit (New England Biolabs, Australia) as per the manufacturer's protocol. Extracted DNA was shipped to Novogene (HK) Co., Ltd for indexing, library preparation, and whole genome sequencing using the NovaSeq 6000 platform. Overall, 128/130 and

2/130 DNA samples were sequenced at expected depths of ~10 G and ~1 G, respectively.

**Bioinformatic analyses**

The bioinformatics workflow of this study is summarised in Supplementary Fig. 12. In addition to the data generated in this study, we included publicly available datasets of individual *D. immitis*. These data were derived from (i) five adults from Sydney, Australia (SRA NCBI: SRR13154013-7)[50]; (ii) one

adult from Georgia, USA (ENA Accession: ERR034940) and one adult from Pavia, Italy (ENA Accession: ERR034941)[51].

## Raw data processing and mapping

Multiple FastQ files were generated for some samples and merged before data analysis. The quality of all raw paired-end sequencing reads was inspected using FastQC (version 0.11.8; https://www.bioinformatics. babraham.ac.uk/projects/fastqc/), and the output was summarised using MultiQC (version 1.17[52]). The raw paired-end reads were processed to remove low quality bases and adapters using *Trim Galore* version (version 0.4.4; https://github.com/FelixKrueger/TrimGalore) with a minimum read length of 50 bp. We used a Nextflow[53] mapping pipeline (mapping-helminth/v1.0.9[54]) to map the trimmed paired-end reads to a combined reference genome containing *D. immitis* (dimmitis_WSI_2.2[25]) and the domestic dog (GenBank accession: GCA_014441545). The dimmitis_WSI_2.2 genome has been previously supplemented with the *D. immitis* mitochondrial and *D. immitis*-associated *Wolbachia* endosymbiont genomes[25]. The mapping pipeline was designed for the standard mapping of helminth genomes using Minimap2 (version 2.17[55]) as a mapper and GATK (version 4.1.4.1[56]), Samtools (version 1.14[57]), and Sambamba (version 1.0[58]) for further data processing. Reads aligning to only the *D. immitis* genome were extracted for downstream analysis using Samtools.

## Sequencing coverage

Sequencing coverage was examined using BamTools (version 2.5.1[59]), BEDTools makewindows (version 2.31.0[60]), and SAMtools bedcov. Coverage statistics were obtained for the nuclear, mitochondrial, and *Wolbachia* genomes of *D. immitis* in 100 kb sliding windows. To visualise nuclear *D. immitis* sequencing coverage for each sample, the scaffold, mitochondrial and *Wolbachia* genomes of *D. immitis* were removed, along with any chromosomes of the domestic dog genome. The ratio of X chromosome to autosome coverage was visualised to determine the sex of the *D. immitis* samples, identified by a drop in coverage along the X chromosome (ratio ~0.5) indicative of a male parasite[25].

## Variant calling

Variants were identified using GATK (version 4.1.4.1[56]) HaplotypeCaller, CombineGVCFs, and GenotypeGVCFs. GVCF files were first produced for each sample and then combined into a cohort-level GVCF file, which was then used for joint genotyping. Outgroup samples were removed to obtain a *D. immitis* cohort VCF file. Variants in the nuclear, mitochondrial, and *Wolbachia* datasets were filtered separately using GATK SelectVariants, GATK VariantFiltration, and VCFtools (version 0.1.16[61]). The distribution of eight quality metrics was established (QUAL, DP, MQ, SOR, QD, FS, MQRankSum, and ReadPosRankSum), and the relevant upper and/or lower 1% distribution tails were removed. The single nucleotide variants were further filtered based on the following parameters: bi-allelic, depth ≥3, and minor allele frequency ≥0.02. Indels were not included in subsequent analyses. Samples with <50% of the total variants were removed, and an optimal threshold for missing data of 0.9 was established (nuclear, mitochondrial, and *Wolbachia*). Additional VCF files were generated from the filtered nuclear data for autosomes (chromosomes 1–4), the sex-linked X chromosome, and Australian samples, as well as VCFs including samples obtained from dog hosts only. SNP discordance in the filtered nuclear autosome data for the technical replicate samples was measured using VCFtools --diff-indv-discordance. All technical replicates had <0.5% SNP discordance with their respective original samples.

## Population structure

To explore the broad-scale genetic relatedness between *D. immitis* individuals and populations, we performed PCA using the R package SNPRelate (version 1.38.0[62]). PCA was conducted for the nuclear (autosomes together, autosomes separate, and chromosome X separate), mitochondrial, and *Wolbachia* data.

Nucleotide diversity (Pi) within populations, as well as absolute nucleotide divergence ($D_{XY}$) and genetic differentiation ($F_{ST}$) between populations, were calculated using pixy (version 2.0.0.beta12[63]). First, the GVCF files were genotyped using GATK GenotypeGVCFs with the inclusion of invariant sites (--all-sites). The nuclear invariants were selected using GATK SelectVariants (--select-type-to-include NO_VARIATION) and merged with the previously filtered nuclear SNPs. The VCF file was filtered to only include invariants and SNPs in the autosomes and sex-linked X chromosome. We conducted the pixy analysis with a window size of 100 kb. Separate runs were performed, grouping the samples in various ways: by region (using dog host samples only), by host species (using all samples), and by city/state (using Australian dog host samples only).

Linkage disequilibrium (LD) statistics were obtained from the nuclear data using VCFtools. First, we filtered the data (--maf 0.02 --max-missing 1). We then calculated the squared correlation coefficient between genotypes (--ld-window-bp 1000000 --max-alleles 2 --min-alleles 2 --geno-r2). To visualise patterns of LD decay, the output was plotted per 0.1 kb for each population.

## Admixture

The global ancestry and admixture proportions of *D. immitis* were inferred using NGSadmix from the ANGSD package[64]. First, genotype likelihoods were extracted from the filtered autosomal *D. immitis* VCF file using VCFtools (--max-missing 1 --BEAGLE-PL). Genotype likelihoods were extracted separately for each chromosome and merged into a single dataset. NGSadmix (-minMaf 0.02 -misTol 0.9) was performed using multiple combinations of different clusters ($K = 2$ to 10) and seeds (1 to 5). CLUMPAK was used to determine the optimal $K$ value[65].

We used TreeMix (version 1.13[66]) to produce a maximum-likelihood tree and infer population splits and admixture events in the *D. immitis* dataset. To generate the input data, VCFtools was used to filter the cohort SNV VCF file (--max-missing 1), retaining only autosomal data. The variants were pruned for linkage disequilibrium (10 kb windows, shifting by 10 kb, LD threshold 0.1), and the VCF file was converted to the appropriate format using custom scripts ('ldPruning.sh' and 'vcf2treemix.sh') obtained from https://github.com/speciationgenomics/scripts. TreeMix was run across a range of migration edges (0 to 5) and seeds (0 to 10), with *D. ursi* included as an outgroup. The optimal number of migration edges was estimated using the R package OptM (version 0.1.8; https://github.com/cran/OptM).

To further investigate admixture in the cohort, *D*-statistics were inferred using AdmixTools (version 8.0.2[67]). First, we converted our filtered nuclear VCF file into the required eigenstrat format using a custom script ('convertVCFtoEigenstrat.sh') obtained from https://github.com/speciationgenomics/scripts/. Additional tools required for file conversion were VCFtools and PLINK (version 1.90b6.18[68]). qpDstat was used to perform *D*-statistic testing for all population pairs relative to the other populations (PopX), with *D. ursi* as the outgroup taxa. *Z*-scores with an absolute value > 3 were considered significant.

## Population demography

We estimated the population size history of *D. immitis* using SMC++ (version 1.15.4[69]). Samples from dog hosts in the filtered autosomal VCF were grouped by continent (i.e. Asia, Australia, Central America, Europe, and USA) and analysed separately before being plotted together. First, a mask of the genome was performed to exclude non-variable sites, using *bcftools query* of the VCF to generate a bed file of variable positions and then *bcftools subtract* using a bed file of the genome to identify genomic positions not found in the VCF. Per population, vcftools was used to extract population-specific variants from the autosomal VCF file (--keep <POP>), ensuring no missing sites (--max-missing 1) and no HWE filter applied. Then, the filtered VCF file was converted to the SMC++ input format using *smc++ vcf2smc* (including --mask missing_sites.bed.gz). *vcf2smc* was run on each chromosome individually ($n = 4$ autosomes), before a combined model was fitted using *smc++ estimate* with a timepoint range of 1 to

1,000,000 and the nematode *Caenorhabditis elegans* mutation rate of $2.7 \times 10^{-9}$ per site per generation as a proxy for the *D. immitis* mutation rate, which is currently unknown. We acknowledge that without a specific *D. immitis* mutation rate, there will be some uncertainty in the population demographic models and that Ne at specific times may be over or under-estimated; however, very few mutation rates have been determined for nematodes[70], and previous studies on parasitic worms have used the *C. elegans* rate as a default[20,21,71,72]. To understand the variance in the model output, we also iteratively ran *smc++ vcf2smc* and *smc++ estimate*, but excluded one chromosome at a time, which allowed us to generate a confidence interval around the "all chromosome" model. To visualise the models, the *smc++ plot* command (-g -c) was used, with the -g parameter specifying the generation time of the target population. The generation time of *D. immitis* is not precisely known, but is estimated to take up to 1 year to produce offspring from an established infection. However, infections are rarely found in hosts <1 year old, and the adult worms can live for up to 7 years in the host's cardiopulmonary system. Furthermore, the development and transmission of this parasite are also highly dependent on local climate and the availability of vectors. To account for potential variability in the generation time of *D. immitis*, we tested multiple generation times (-g 1, 2.5, or 4) (Supplementary Fig. 13), settling on a midpoint of 2.5 years for major comparisons.

## Statistics and reproducibility

Previously reported adult heartworm burdens in various mammalian hosts were obtained from existing literature (Supplementary Data 2). These hosts included dogs, wolves, coyotes, jackals, foxes, cats, raccoons, badgers/ferrets, seals, and bears. To test for significant differences in worm burdens between these hosts, we conducted Kruskal–Wallis tests with Dunn's multiple comparisons using GraphPad Prism (version 10.1.2). In addition, we performed Wilcoxon Rank Sum tests in R (version 4.4.1) to test for differences in Pi between continents in our heartworm cohort.

To promote reproducibility, the raw sequencing data and code used in this study have been made publicly available (see *Data availability* and *Code availability* below). To confirm that the genetic patterns we observed were in fact biological and not artifactual, we included five technical replicates from multiple geographical locations (Brisbane, Australia: $n = 1$; Sydney, Australia: $n = 2$; Malaysia: $n = 1$; Panama: $n = 1$). For these replicates, we extracted and sequenced DNA from a second segment of the original worm. All replicates clustered in the same positions as their corresponding originals in the PCA plots, with the exception of 'MYS_SEL_AD_001_R' which was excluded from the nuclear PCA due to low coverage.

## Reporting summary

Further information on research design is available in the Nature Portfolio Reporting Summary linked to this article.

## Data availability

Raw FastQ sequences generated in this study are available at SRA NCBI BioProject PRJNA1104412. Source data underlying the main figures are provided in Supplementary Data 4 where appropriate, with additional genomics-specific files in Zenodo[73].

## Code availability

The code for this study is available from GitHub (https://github.com/rosemondepower/heartworm_genome.git) and is archived in Zenodo[73].

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

## Acknowledgements

The authors gratefully acknowledge Julie Austin, Iris Cheadle, Constantin Constantinoiu, Carol Esson, Eloise Fenton, Linda Hayes, Tessa Mackie, Yissu Martinez, Sandi McClintock, Ilze Nel, Tony Phillis, Duncan Smith, and Jim Taylor for their generous provision of heartworm specimens for this study. Sampling from the University of Queensland was supported by the University of Queensland, School of Veterinary Science Bequest and Donation Funds for Research (Stone Memorial Fund). We thank Javier Gandasegui for their foundational bioinformatics work that facilitated this work. We acknowledge the Pathogen Informatics group at the Wellcome Sanger Institute and the Sydney Informatics Hub for bioinformatics support. RIP was supported by an Australian Government Research Training Program Stipend and The Jean Walker Trust Fellowship. RIP's collaborative research exchange was supported by the University of Sydney's Eric Horatio Scholarship, the University of Sydney's Postgraduate Research Support Scheme, and the Australian Society for Parasitology's Researcher Exchange, Training and Travel Award. SRD is supported by a UKRI Future Leaders Fellowship [MR/T020733/1] and the Wellcome Trust (UK) through core funding to the Wellcome Sanger Institute (UK) [220540/Z/20/A]. This work was funded by the Canine Research Foundation, Dogs Victoria, Australia and the Australian Companion Animal Health Foundation Research Grant.

## Author contributions

R.I.P. and J.Š. conceived and designed the study; J.Š. and S.R.D. co-supervised the study; S.A., H.S.W., G.G.V., T.L.S., J.L.L., A.R.M., M.T.D., J.M.F., M.L.M., A.R., S.C.C., G.D., A.D.M., P.D., E.P., P.T., D.T.B., A.D.N., R.S.K.S., and J.Š. provided materials; R.I.P. extracted DNA and prepared samples for sequencing; R.I.P. and S.R.D. led and performed the bioinformatics analyses; R.I.P. and S.R.D. analysed and interpreted the results, with input from J.Š. and S.Y.W.H.; R.I.P. drafted the original manuscript, with contributions from S.R.D. and J.Š. All authors contributed to the revision and approved of the final manuscript.

## Competing interests

The authors declare no competing interests.

## Additional information

[1]Sydney School of Veterinary Science, Faculty of Science, The University of Sydney, Sydney, NSW, Australia. [2]The University of Queensland, School of Veterinary Science, Gatton, QLD, Australia. [3]Department of Comparative, Diagnostic, and Population Medicine, College of Veterinary Medicine, University of Florida, Gainesville, FL, USA. [4]Department of Veterinary Pathobiology, College of Veterinary Medicine & Biomedical Sciences, Texas A&M University, College Station, TX, USA. [5]Department of Clinical Sciences, College of Veterinary Medicine, North Carolina State University, Raleigh, NC, USA. [6]Department of Infectious Diseases, College of Veterinary Medicine, University of Georgia, Athens, GA, USA. [7]New England Biolabs Inc., Ipswich, MA, USA. [8]University of Wisconsin Oshkosh, Oshkosh, WI, USA. [9]Laboratory of Helminthology, Faculty of Microbiology, University of Costa Rica, San José, Costa Rica. [10]Centro de Investigación en Enfermedades Tropicales, University of Costa Rica, San José, Costa Rica. [11]Department of Veterinary Parasitology, Veterinary Sciences, Healthy Pet Veterinary Hospital SC, David, Chiriquí, Panamá. [12]Department of Parasitology and Parasitic Diseases, University of Agricultural Sciences and Veterinary Medicine of Cluj-Napoca, Cluj-Napoca, Romania. [13]SCS3 - Parasitology and Mycology Unit, Istituto Zooprofilattico Sperimentale delle Venezie, Legnaro, Italy. [14]Laboratory of Parasitology and Parasitic Diseases, Faculty of Veterinary Medicine, Aristotle University of Thessaloniki, Thessaloniki, Greece. [15]Parasitology Unit, Department of Veterinary Pathology, Faculty of Veterinary Science, Chulalongkorn University, Bangkok, Thailand. [16]Biomarkers in Animals Parasitology Research Unit, Chulalongkorn University, Bangkok, Thailand. [17]Institute of Biology, Vietnam Academy of Science and Technology, Hanoi, Vietnam. [18]Faculty of Biology, Graduate University of Science and Technology, Vietnam Academy of Science and Technology, Hanoi, Vietnam. [19]Department of Medical Parasitology, Vietnam Military Medical University, Hanoi, Vietnam. [20]Faculty of Veterinary Medicine, Universiti Putra Malaysia, Serdang, Selangor, Malaysia. [21]School of Life and Environmental Sciences, Faculty of Science, The University of Sydney, Sydney, NSW, Australia. [22]Wellcome Sanger Institute, Hinxton, Cambridgeshire, UK. [23]Sydney Infectious Diseases Institute, The University of Sydney, Sydney, NSW, Australia. [24]Present address: Centre for Palaeogenetics, Stockholm, Sweden. [25]Present address: Department of Zoology, Stockholm University, Stockholm, Sweden. [26]These authors jointly supervised this work: Stephen R. Doyle, Jan Šlapeta. ✉e-mail: stephen.doyle@sanger.ac.uk; jan.slapeta@sydney.edu.au

