## [Transparent Peer Review file · Communications Biology]

Population genomics reveals an ancient origin of heartworms in canids

Corresponding Author: Professor Jan Šlapeta

Version 0:

Reviewer comments:

Reviewer #1

(Remarks to the Author)

The authors present a population genomics study of the nematode heartworm *Dirofilaria immitis*, a common parasite of dogs and other canids. The sampling across the world is quite good, and sequencing coverage very high. They perform various standard population genomics analyses, all of which appear to be appropriately performed. They address various questions around the diversity and history of the heartworm, in particular the timing of diversification in the species and how this aligns with dog and human history.

Key to most of the conclusions of the paper are the genetic divergence time inferences. These are made using the *smc++* software (Fig 3a). This software has to date primarily been used for effective population size inference, and while it has the ability to infer split times too, it is less tried and trusted for this purpose. One aspect is that the *smc++* split time inference assumes a clean split, with no gene flow after initial separation, an assumption which might or might not apply well in any particular species. There are also no confidence intervals provided with the point estimates, which makes interpretation trickier. I have concerns that the split times might not be so accurate – in particular that they are too old - and that this might affect the various conclusions the authors draw.

The authors estimate a divergence between Asian and Australian heartworms of ~29 ky, thus greatly predating the arrival of the dingo to Australia. They attribute this to the Asian source being unsampled, and there potentially being substantial divergence between the unsampled source (perhaps somewhere in Indonesia) and the Asian genomes they have sampled here (Thailand and Malaysia). This is certainly possible, but it is still a surprisingly deep split time. Perhaps they would get a more recent divergence if they had samples from Indonesia, but it's worth keeping in mind that dogs have not been in Indonesia for very long – they probably only arrived in Indonesia shortly before arriving in Australia. And there are very few wild canids in Indonesia, only the dhole – no foxes, wolves or jackals. So actually, dingoes probably had a common ancestor with dogs in Thailand and Malaysia something like 5-10 kya.

It's hard to take the inferred split time of 29 kya as evidence that the heartworm came to Australia with dingoes ~5 kya. With that kind of mismatch, it seems like an arrival from modern European or Asian dogs in the period of global trade after European colonisation is a pretty much equally plausible hypothesis. Neither hypothesis explains the inferred split time.

The authors propose a very recent split between European and Central American heartworms. But I can't find this divergence time estimate – it's not listed in Fig 3a – unless I have missed it somewhere? If *smc++* actually estimates a split time of just a few hundred years here, then that would be quite reassuring and show that method is able to infer a recent split. That would increase confidence in the general approach and thus all the other split times too. But if *smc++* infers a European-Central American split of >20 kya like all the other split times, then this would not be reassuring.

The finding that heartworms from non-dog species all cluster by geography rather than species is a very interesting one, and demonstrates that there must be frequent and local cross-species transmission.

Based on the very deep split times inferred overall, some as deep as 46-63 kya, the authors propose that heartworm diversification substantially preceded dog domestication, with the implication that it must have spread among wild canids. This is certainly possible, and would be an important finding if correct, but given my doubts about the split time inferences I am not fully convinced about this claim. Especially given the apparently frequent cross-species transmissions, and

widespread recent movement of dogs, it seems a little bit difficult to believe such deep divergences. The authors discuss glaciation as a possible driver, but this is a quite vague and general discussion. Even within Eurasia there is a very deep split inferred between Asia and Europe (46 kya). In the last 5000 years there has been very extensive movement of dogs between Europe and Asia – e.g. most dogs in China today, even village dogs, have some amount of European dog ancestry. It seems difficult to believe that the heartworm populations would have remained isolated for so long. Not necessarily impossible – but the results presented here do not really convince me, at least.

The claim of “Bering Land Bridge allowed panmictic heartworm populations across northern hemisphere” (Fig 6a) is little bit odd, and at odds with the claim of deep structure in heartworms. If the heartworm population was panmictic across the Bering Land Bridge, we would expect a Eurasia-North America split time of more like ~15 kya, which is when the bridge was last opened. In general I don’t think the authors have any evidence to demonstrate panmixia in Pleistocene heartworms.

I think what would strengthen the results and conclusions would be some orthogonal evidence on the timing of diversification. This could be e.g. estimates with some other software such as MSMC (e.g. run on haploid X male chromosomes, to get around the need for phasing) or other, and/or evidence from the mitochondrial genomes, where divergences are easier to calculate. Do the mitochondrial divergences support early splits? I note that the authors found only 32 mitochondrial SNPs, which seems like a quite small number, and could potentially reflect recent mitochondrial divergences.

I think the Methods description of how smc++ is run is not quite satisfactory. smc++ requires some care to run, as it does not simply make use of variant sites like PCA or ADMIXTURE. Instead it also makes use of non-variable sites. It is thus crucial that non-variable sites are distinguished from sites that are simply not called. The Methods section does not explain how this is done. Were non-variable positions explicitly output from GATK, and used for the smc++ inference? Was a genome mappability mask applied? Inappropriate treatment of non-variable sites could result in inaccurate population size and split time results, so it’s important to deal with this correctly, and to describe what was done in detail.

Another question is, how accurate is the mutation rate? It is taken from *C. elegans*, but could the true mutation rate be higher? If so, that would cause split times to be over-estimated (assuming a mutation rate that is too low means that a longer time is needed to accumulate a given amount of divergence).

A filter of “Hardy-Weinberg Equilibrium p-value $\geq 10^{-6}$ ” was applied to the SNPs. In every structured population, many variants will depart from HWE because of population structure – there will be too many homozygotes relative to HWE, because alleles cluster in space and often show up in the same individuals. It is thus arguably ideal to filter not on departure from HWE (which can be caused by either Excess Heterozygosity or Excess Homozygosity), but only on Excess Heterozygosity. Though I would not imagine that this has had any substantial impact on the analyses. But it’s also the kind of filtering that is not suitable for smc++, because it’s a filter that applies to variable sites but not non-variable sites.

“Dingoes are an ancient dog breed from Asia” – I don’t think dingoes should be described as a “breed”.

Reviewer #2

(Remarks to the Author)

Slapeta and coworkers report a whole genome comparison of over 100 adult *Dirofilaria immitis* with the goal to investigate the evolutionary history of heartworms and to test the common hypotheses regarding the anthropocentric radiation of heartworms. In a principle component analysis they find four clusters, corresponding to the four continents where samples were collected. Interestingly, samples from Europe and Central America, although still discriminable, clustered together. Distinct genetic profiles could be observed for all four groups, suggesting a restricted spatial spread in the history of the species.

Samples of heartworms collected from non-canids allowed for the assessment of the genetic diversity between host organisms. While such diversity can be observed, it was postulated that geography rather than host species dominated the genetic variation.

Population split time analyses revealed that the divergence between populations is older than the proposed dates for dog domestication, implying that the spread of heartworms is not related to the event of dog domestication. Further analysis suggests canids like wolves or coyotes as the most likely primary hosts for heartworms in ancient times.

Co-evolution of Heartworms and canids is suggested since the Oligocene. The spread to Eurasia and South America is explained with the formation of the Bering land bridge and the Isthmus of Panama. At this time heartworms were still one population. Divergence occurred at the time of the last interglacial period. The ice sheets of the following cool period separated the ancient canid populations, and with them, potentially also the heartworm populations.

Heartworm populations in Australia and Asia are related and it was postulated that the Australian Heartworm population descends from Asian populations. Introduction of heartworms to Australia is more likely to dingo hosts than to an introduction through European colonisation. A discrepancy between the estimated divergence of Asian and Australian heartworms is explained with a geographically limited sample set of Asian heartworms. Additional samples from broader geographical regions are needed.

The co-clustering of European and Central American populations might be due to canid introduction during the colonisation of the Americas in the late 15th century. Split times support this hypothesis, rather than an introduction during recent migrations between Europe and Central America.

The manuscript is well written and follows a logical argumentation line. The topic is of interest to the scientific community and

helps to understand the relation between different heartworm populations, which can be of interest also in the development of future heartworm preventatives and therapeutics. The data are available as well as the code used in the analysis. All programs and software suites used are described in the Methods section, allowing for the reproduction of the work.

I recommend to publish the manuscript with a minor modification (see below).

Lines 78-80: "This is of particular concern for domestic dogs, whereby potentially life-threatening cardiopulmonary disease is globally managed by lifelong and rigorous application of parasiticides 3." Although a treatment option for infected dogs exist, the main means is prevention using macrocyclic lactones. The therapeutic option includes the application of melasormine, inevitably leading to thromboembolism (https://d3ft8sckhnqim2.cloudfront.net/images/AHS_Canine_Guidelinesweb22NOV2024.pdf?1732318144). I suggest to rephrase the sentence in lines 78-80 to emphasize the prevention instead of the treatment.

Richard J. Marhöfer

Reviewer #3

(Remarks to the Author)

This is a very nice and important population genomic study which provides a new hypothesis-based framework for global population dynamics and evolutionary history of the canine heartworm *Dirofilaria immitis*. This is the largest and most thorough study of its type to date being based on whole genome sequence data from 127 individual adult worms from different carnivore hosts across four continents. The sequence data and population genomic analysis is thorough and clearly presented and the results are interesting and important.

The conclusions are very interesting suggesting a much deeper evolutionary history of *D. immitis* resulting in significant contemporary geographical sub-structuring at the continental level and supporting the hypothesis that the parasite first evolved in canids on the North American continent. Subsequent spread into Eurasia with wild canid migration across the Bering land bridge led to panmictic populations across the northern hemisphere pre-glaciation before icesheets during glaciation periods led to separation of population and genetic drift. Further hypotheses regarding subsequent spread into Australasia from Asia and a much more recent reintroduction and admixture into central America through European settlement are suggested. I am not an expert on the evolutionary analysis aspects but these hypotheses are clearly presented and seem to be well argued with appropriate caveats.

I think this paper is a good fit for Communications Biology and I only have a few minor critical comments:

Line 154 – “sample” should read “sampling”

Lines 182-184- The data suggesting “suggesting that heartworms from canids and mustelids may be more genetically similar to each other than to those from felids” doesn’t seem that strong to me. In fact looking at extended data figure 5b the fox samples are the only ones that stand out with moderate F_{st} s against most other hosts (except domestic dog). I would be careful not to make the case too strongly that feline *D. immitis* may be genetically different from canine *D. immitis* as per statement cited above as this would have important implications and I don’t think the data is strong enough to support it. Lines 62-64- I don’t think the overall hypothesis is well captured in the abstract –after the statement “Population structure and demographic analyses of the nuclear genome reveal distinct genetic differences between heartworms from different continents, indicating a deeper ancient origin and dispersal in canid hosts than previously recognised” the migration into Australia with dingoes and the more recent re-introduction into central America by early European colonizers is discussed, instead, it would be helpful to have an additional sentence or two to explain the overarching hypothesis that heartworm originated in the Americas and was panmictic with Eurasia pre-glaciation until populations were separated during glacial periods and allowed genetic separation.

Also, the final phrase of the final sentence of the abstract is “..which can aid future surveillance and control efforts for this important veterinary parasite”. This seems very vague and is not explained in the discussion. I suggest it is either removed or a paragraph in the discussion explaining why this work is relevant to surveillance and control efforts would be needed. eg. that molecular diagnostic tests need to account the high level of genetic differentiation globally and for the fact that genetic markers of anthelmintic resistance may not be universal applicable globally.

Version 1:

Reviewer comments:

Reviewer #1

(Remarks to the Author)

The authors have substantially revised the part of the paper I was most sceptical about, the divergence times estimates using $smc++$. They performed some additional analyses and felt that these were not very reliable, and should be commended on taking the decision to leave this out, despite the large amount of work that will have gone into it. They also have revisited how they run the software for population size inferences, which now appears to be appropriate. Many of my technical concerns on this are thus now resolved. I still feel like the authors are a bit over-confident in their conclusions about the history of the heartworm, are overinterpreting the detailed shapes of N_e curves, and are perhaps overfitting their prior expectations a bit to the results. I would welcome some softening of the conclusions and acknowledgement of caveats. I

offer various feedback in the below, but would overall say that I think it's a good paper that should be published, and I don't necessarily think another round of review should be necessary.

The authors now take a different approach to estimating divergence times: they look at when effective population (N_e) size history curves start to diverge from each other. This approach was perhaps taken early in the days of PSMC, but is not so common. It's a very indirect approach. The obvious potential limitation is that two populations sometimes might retain similar population sizes after their split – you will then not get any visible divergence in the N_e curves, despite a split having occurred. There is thus always a risk of underestimating the split time. On the other hand, if a divergence of the curves is observed, it is likely this reflects some degree of genetic separation.

It is in some sense a little bit problematic that, after revising their divergence time estimates quite substantially (e.g. 2-fold differences compared to first submission in some cases), the authors don't really change their conclusions. E.g. a divergence time of 29 kya and a divergence time of 10-20 kya are both taken as evidence of introduction to Australia with dingos. I think this shows the importance of prior hypotheses in shaping conclusions. This being said, the new version of the paper is probably more likely to align with the truth.

“Considering the variation in population demographic curves >40 kya (Figure 3a), we propose that there were substantial differences in some populations that were evolving independently before dog domestication”. There is some variation before >40 kya, but at least to me it's not clear that this is not just noise. All curves show a highly correlated pattern, first going down, and then up quite dramatically around ~50 kya - but are deviating a little bit in these periods. I would guess that degree of deviation could probably emerge as noise with smc++. Single-genome PSMC or MSMC curves are likely more robust in the distant past. As an example, there is quite some difference >40 kya between the European and Central America (CENAM) curves. The authors otherwise believe that CENAM is a very recently derived offshoot from Europe, in which case their curves should be identical >40 kya. Do the authors really believe that the Europe and CENAM curves reflect real difference in that time period? If not, I would not in general interpret these ancient differences as reflecting independent evolution in those time periods. Personally, if I just look at Figure 3a my take-away would probably be that genuine divergences only really start at 10 kya.

The authors say that between Europe and CENAM “there is a striking divergence in N_e between populations around 500 years ago (Figure 3d)”. Looking at Figure 3d, I would not describe this as a striking divergence. The two curves stay at very similar levels from then until the present day. There is always some minor fluctuations in these curves, but I would not interpret these results as clear evidence of genetic divergence, certainly not “striking”. It's also not necessarily clear if smc++ has resolution in this recent time frame. I would focus on the genetic clustering results as the basis of the evidence for a recent European origin.

The authors cite an entry time for the dingo in Australia of 5-12 kya, citing dog genetic analyses – but genetic divergence times are not the same thing as entry times. 5-12 kya would be really quite early, not really aligning with the archaeological record. The earliest dog remains in Australia are from ~3.5 kya, and many would think an entry time of around 4 kya is likely. This would also align with the history of Austronesian people migrating down from Southeast Asia. The authors' new Asian-Australian heartworm divergence estimate of “10–20 kya” does still not straightforwardly align with dingo entry at 4 kya. I thus feel the conclusions on this point are a bit too strong.

The alternative hypothesis on Australia, which to me would also seem about equally compatible with the data, is that heartworms came to Australia in the post-colonial era, from some domestic dogs brought in from Southeast Asia, China, or other parts of Asia. E.g. there are many Asian dog breeds kept as pets in Australia today, and there would have been plenty of trade and opportunity for recent introductions from different parts of Asia over the past couple of centuries. If the recent source is also from some unsampled region – for example, let's say Japan, or northern China – there would still be some amount of genetic divergence observed between the Australian heartworms and the sampled Asian heartworms in Thailand.

The discussion on the question “If heartworms were once one population, when did they diverge, and why” is a little bit confusing to me. It mainly talks about correlated changes in N_e curves – which typically is interpreted as shared ancestry and history. It talks about glaciation potentially fragmenting heartworm populations, but this is not really seen in diverging N_e curves. The authors refer to “this fragmentation”, but fragmentation is only really clearly seen after 10,000 kya.

“This time period [58–72 kya] was associated with a fall in N_e , particularly in the USA samples” – this is pretty much when the N_e curves peak to their maximal values, so slightly odd conclusion to take this as evidence for glaciation-induced declines. Overall, this section feels a bit too much like just eyeballing the N_e curves and then engaging in storytelling of a general narrative connecting the obtained results to whatever climate changes happened to occur in the time periods. I similarly think the claim of “multiple waves of dispersal with canids across the Bering Land Bridge” – I simply think this level of detail is not possible to infer just by looking at these N_e curves.

The mitochondrial results are a little bit puzzling, as the authors say they observe very little variation in the mitochondrial genome. This would imply a very recent common ancestor of all heartworm mitochondria.

Reviewer #3

(Remarks to the Author)

I am satisfied with the responses to my comments and the edits to the manuscript in the revised version.

Version 2:

Reviewer comments:

Reviewer #1

(Remarks to the Author)

I think the more careful caveating in response to reviewer comments has benefitted the paper. I don't think further review is needed. I just leave the authors with the below two points, where I think slight tweaks would be appropriate.

I appreciate how the authors acknowledge that post-colonial introduction of the heartworm to Australia is also consistent with the data. However, in the abstract they still only favour the dingo hypothesis, which I thus don't think is appropriately representing the revised paper. Also, in the main text, even though they now describe the post-colonial introduction as "perhaps equally plausible", higher up they say "... dingo hosts, rather than through introduction by European colonisation".

In the response letter, the authors note the surprisingly low amount of diversity in the mitochondrial genome, "but at this moment we refrain from making any speculation about the reasons or causes of it." I think it's fine to not be able to explain everything, but from one point of view this is not ideal: one line of evidence does not align with the hypothesis of ancient diversification, so the authors do not even mention that evidence. An alternative hypothesis is that heartworm diversification is only a few hundred years deep, and the mitochondrial data reflects that, and then the deeper timescales inferred from the autosomal data are artefacts of the complicated smc++ method. I don't necessarily think that's the case, but I think it would be appropriate to mention with at least one sentence that the low amount of diversity in the mitochondrial genome is surprising given the proposed ancient diversification, and that the authors currently do not have an explanation for this.

Dear Editors,

RE: revision of “Population genomics reveals an ancient origin of heartworms in canids” (COMMSBIO-25-1196-T)

We would like to thank the editorial team and the three reviewers for their careful consideration and positive overall appraisal of our manuscript. We appreciate the constructive comments and suggestions, which we have addressed in full below, that have helped us clarify the presentation of our data and conclusions. We have included marked-up and clean versions of the revised manuscript to highlight the changes made.

We hope these revisions are sufficient for you to reconsider publishing our manuscript in Communications Biology.

Kind regards,

Jan Slapeta and Stephen R. Doyle
Co-corresponding authors, on behalf of all authors.

RE: revision of “Population genomics reveals an ancient origin of heartworms in canids” (COMMSBIO-25-1196-T)

Response to reviewers

Referee expertise:

Referee #1: population genomics, canid genetics, evolution

Referee #2: parasitology, computational biology

Referee #3: parasitology, genomics

Reviewers' comments:

Reviewer #1 (Remarks to the Author):

The authors present a population genomics study of the nematode heartworm *Dirofilaria immitis*, a common parasite of dogs and other canids. The sampling across the world is quite good, and sequencing coverage very high. They perform various standard population genomics analyses, all of which appear to be appropriately performed. They address various questions around the diversity and history of the heartworm, in particular the timing of diversification in the species and how this aligns with dog and human history.

Key to most of the conclusions of the paper are the genetic divergence time inferences. These are made using the smc++ software (Fig 3a). This software has to date primarily been used for effective population size inference, and while it has the ability to infer split times too, it is less tried and trusted for this purpose. One aspect is that the smc++ split time inference assumes a clean split, with no gene flow after initial separation, an assumption which might or might not apply well in any particular species. There are also no confidence intervals provided with the point estimates, which makes interpretation trickier. I have concerns that the split times might not be so accurate – in particular that they are too old - and that this might affect the various conclusions the authors draw.

RESPONSE: We appreciate the reviewers' comments and concerns regarding the divergence time estimates, particularly around the use of smc++ split to infer when two populations diverge after a shared coevolving history. Further, we acknowledge that a concern regarding the use of smc++ highlighted by the reviewer....

“smc++ split time inference assumes a clean split, with no gene flow after initial separation, an assumption which might nor might not apply well in any particular species”

...is likely valid here. In fact, some of our inferences specifically reference secondary contact between populations, for example, between the Asian and the USA populations. Hence, we acknowledge some issues here.

In response to this comment and further reviewer comments below, we completely reran our smc++ analyses, incorporating genome masking and jackknifing (as suggested below), before rerunning smc++ split. First, we believe the reviewers' suggestions have improved the estimates of Ne change for all populations, and that they make more sense in terms of showing consistent trends that might be expected by shared demographic histories, as well as more consistent contemporary Ne, which now correlate well with estimates of nucleotide diversity. However, regarding estimates of split times, we found them to be equally inconsistent. Moreover, after a close examination of all pairwise comparisons (pop1 vs. pop2) and reciprocal comparisons (pop2 vs. pop1), we observe very different split times that appear to be highly dependent on stochastic differences in the demographic curves. Revising the original analyses, we believe the same effect is occurring. Therefore, we are not confident in the split time estimates from smc++ split, in both the original and reanalysed data.

In response to this comment and reanalysis, we have decided to remove all inferences of specific split times and instead provide more explicit pairwise comparisons of Ne to explore specific hypotheses in our data, making it clearer when we speculate about whether the data support our hypothesis. Overall, we are confident in the general trends of the reanalysed data; however, we are now much more aware of the limitations of using smc++ split to infer specific demographic change.

The authors estimate a divergence between Asian and Australian heartworms of ~29 ky, thus greatly predating the arrival of the dingo to Australia. They attribute this to the Asian source being unsampled, and there potentially being substantial divergence between the unsampled source (perhaps somewhere in Indonesia) and the Asian genomes they have sampled here (Thailand and Malaysia). This is certainly possible, but it is still a surprisingly deep split time. Perhaps they would get a more recent divergence if they had samples from Indonesia, but it's worth keeping in mind that dogs have not been in Indonesia for very long – they probably only arrived in Indonesia shortly before arriving in

Australia. And there are very few wild canids in Indonesia, only the dhole – no foxes, wolves or jackals. So actually, dingoes probably had a common ancestor with dogs in Thailand and Malaysia something like 5-10 kya.

It's hard to take the inferred split time of 29 kya as evidence that the heartworm came to Australia with dingoes ~5 kya. With that kind of mismatch, it seems like an arrival from modern European or Asian dogs in the period of global trade after European colonisation is a pretty much equally plausible hypothesis. Neither hypothesis explains the inferred split time.

RESPONSE: As outlined above, we have removed the specific split time.

We respectfully disagree that a post-colonial introduction, especially for European dogs, is equally plausible, and to some extent, also for Asian dogs (although the sampling is extremely limited). Our genomic data show no evidence of recent admixture or introduction from European sources. If heartworm had arrived with imported dogs in the colonial period, we would expect to see genetic signatures of those lineages, which are absent. Moreover, successful establishment requires not just an infected host but also a competent mosquito vector. Not all mosquito species can transmit heartworm effectively. Thus, even if infected dogs arrived post-colonially, without the right vector, transmission would not occur, making such introductions evolutionary dead ends. We are aware of dholes that are a potential host, but obtaining samples remains challenging.

However, in our new analysis of the smc++ data, we do show divergence in AUS and ASIA population Ne around 5-12 kya in line with the dingo introduction into Australia, followed by a progressive increase in Ne in Australia only. This pairwise comparison is shown in the new figure panel, Figure 3c.

The authors propose a very recent split between European and Central American heartworms. But I can't find this divergence time estimate – it's not listed in Fig 3a – unless I have missed it somewhere? If smc++ actually estimates a split time of just a few hundred years here, then that would be quite reassuring and show that method is able to infer a recent split. That would increase confidence in the general approach and thus all the other split times too. But if smc++ infers a European-Central American split of >20 kya like all the other split times, then this would not be reassuring.

RESPONSE: As outlined above, we have removed the specific split time.

However, in our new analysis of the smc++ data, we do show increased variance around 500 yrs after what looks to be a fairly correlated N_e in the past, consistent with our hypothesis of a recent split. This pairwise comparison is shown in the new figure panel Figure 3d.

The finding that heartworms from non-dog species all cluster by geography rather than species is a very interesting one, and demonstrates that there must be frequent and local cross-species transmission.

RESPONSE: Yes, we agree that it is interesting and that there must be frequent transmission. Unfortunately, we don't have broader sampling to explore this further, as our non-dog samples were limited and geographically restricted. However, this is clearly demonstrated through more in-depth sampling across Australia of dogs and foxes.

Based on the very deep split times inferred overall, some as deep as 46-63 kya, the authors propose that heartworm diversification substantially preceded dog domestication, with the implication that it must have spread among wild canids. This is certainly possible, and would be an important finding if correct, but given my doubts about the split time inferences I am not fully convinced about this claim. Especially given the apparently frequent cross-species transmissions, and widespread recent movement of dogs, it seems a little bit difficult to believe such deep divergences. The authors discuss glaciation as a possible driver, but this is a quite vague and general discussion. Even within Eurasia there is a very deep split inferred between Asia and Europe (46 kya). In the last 5000 years there has been very extensive movement of dogs between Europe and Asia – e.g. most dogs in China today, even village dogs, have some amount of European dog ancestry. It seems difficult to believe that the heartworm populations would have remained isolated for so long. Not necessarily impossible – but the results presented here do not really convince me, at least.

RESPONSE: We appreciate the reviewer's scepticism and agree that the deep divergence times warrant careful interpretation. However, we emphasise that heartworm transmission depends not only on the presence of canid hosts but also on suitable mosquito vectors and environmental conditions: specifically, regions with >130 days annually above 15°C to support larval development. While canids are highly mobile and cross-species transmission is common, the

parasite's life cycle imposes ecological constraints that can maintain population structure over long timescales. Thus, even with extensive dog movement in recent times, vector and climate limitations could have restricted gene flow between heartworm populations, preserving deep divergences. We acknowledge that further sampling and modelling are needed to refine these estimates, but the current data support the possibility of long-term isolation shaped by ecological, not just host-related, factors.

The claim of "Bering Land Bridge allowed panmictic heartworm populations across northern hemisphere" (Fig 6a) is little bit odd, and at odds with the claim of deep structure in heartworms. If the heartworm population was panmictic across the Bering Land Bridge, we would expect a Eurasia-North America split time of more like ~15 kya, which is when the bridge was last opened. In general I don't think the authors have any evidence to demonstrate panmixia in Pleistocene heartworms.

RESPONSE: As above, we have downplayed inferences of specific split times.

We looked at the ASIA / USA divergence in the new smc++ analyses, and while speculative, there is evidence of N_e converging around 15 kya. Older than this time, N_e of both ASIA and USA seem to diverge; they show consistent patterns overall, but they are offset. This may be technical or biological; however, we can't be sure. We show this pairwise comparison in the new figure panel Figure 3b, highlighting potential time points (11-30 kya and 60-70 kya) during which the Bering land bridges are proposed to be open.

I think what would strengthen the results and conclusions would be some orthogonal evidence on the timing of diversification. This could be e.g. estimates with some other software such as MSMC (e.g. run on haploid X male chromosomes, to get around the need for phasing) or other, and/or evidence from the mitochondrial genomes, where divergences are easier to calculate. Do the mitochondrial divergences support early splits? I note that the authors found only 32 mitochondrial SNPs, which seems like a quite small number, and could potentially reflect recent mitochondrial divergences.

RESPONSE: As outlined above, we have downplayed specific split times due to our improved understanding of the variance and uncertainty associated with these split times.

Regarding the approach, MSMC can only handle a maximum of eight samples, whereas smc++ can handle larger sample sizes and does well with unphased data, and is the preferred approach when analysing large numbers of genomes.

Surprisingly, the mtDNA variation proved uninformative in terms of population structure, as shown in Supplementary Figure 7a, and therefore would not help in understanding split times.

I think the Methods description of how smc++ is run is not quite satisfactory. smc++ requires some care to run, as it does not simply make use of variant sites like PCA or ADMIXTURE. Instead it also makes use of non-variable sites. It is thus crucial that non-variable sites are distinguished from sites that are simply not called. The Methods section does not explain how this is done. Were non-variable positions explicitly output from GATK, and used for the smc++ inference? Was a genome mappability mask applied? Inappropriate treatment of non-variable sites could result in inaccurate population size and split time results, so it's important to deal with this correctly, and to describe what was done in detail.

RESPONSE: We agree that we could have been more descriptive in our description of the smc++ methods. However, this comment prompted us to reflect on our approach and led us to completely rerun smc++. In doing so, we:

(i) generated a genome mask to exclude non-variable sites. There was considerable variation in the literature on the best approach to doing this, from masking everything except variable sites, to creating a "mappability mask" on predicted regions in which reads map uniquely (following Heng Li's SNPability code), to masking repeats from RepeatMasker. Surprisingly, even the authors of smc++ recommended masking (but only as an optional, not mandatory parameter), without specifying a particular approach. We tested and compared a full mask, a mappability mask, and no mask (original analyses). The mappability mask and no-mask analyses produced similar outputs, and hence, led us to conclude that partial mapping was insufficient. However, we found that a full mask provided the least noise and generated N_e values that were more consistent overall with the expected timing of divergence and contemporary nucleotide diversity levels.

(ii) In addition to rerunning smc++ on all chromosomes, we performed jackknifing by iteratively dropping one chromosome from the analysis, which allowed us to better understand the variance in the estimates.

These changes are now reflected in the figures and are fully described in the results and methods sections, where applicable.

Another question is, how accurate is the mutation rate? It is taken from *C. elegans*, but could the true mutation rate be higher? If so, that would cause split times to be over-estimated (assuming a mutation rate that is too low means that a longer time is needed to accumulate a given amount of divergence).

RESPONSE: This is a really good question, and the answer is that we do not know how accurate using a *C. elegans* mutation rate for *D. immitis*. *C. elegans*, the model freeliving nematode, is one of only a few nematodes for which a mutation rate has been determined via mutation accumulation experiments. Moreover, the *C. elegans* mutation rate is quite similar to two other freeliving nematodes (Wang and Obbard 2023; <https://doi.org/10.1093/evlett/qrad027>). Determining the mutation rate for a parasitic nematode like *D. immitis* would be extremely difficult. Hence, we, along with other studies on parasitic nematodes, have relied on the *C. elegans* mutation rate.

In response to this comment, we have reiterated that, in the absence of specific *D. immitis* data, there is uncertainty in using the *C. elegans* mutation rate.

“vcf2smc was run on each chromosome individually (n = 4 autosomes), before a combined model was fitted using smc++ estimate with a timepoint range of 1 to 1,000,000 and the nematode *Caenorhabditis elegans* mutation rate of 2.7×10^{-9} per site per generation as a proxy for the *D. immitis* mutation rate, which is currently unknown. We acknowledge that without a specific *D. immitis* mutation rate, there will be some uncertainty in the population demographic models and that Ne at specific times may be over or underestimated; however, very few mutation rates have been determined for nematodes⁶³, and previous studies on parasitic worms have used the *C. elegans* rate as a default^{16,17,64,65}.”

A filter of “Hardy-Weinberg Equilibrium p-value $\geq 10^{-6}$ ” was applied to the SNPs. In every structured population, many variants will depart from HWE because of population structure – there will be too many homozygotes relative to HWE, because alleles cluster in space and often show up in the same individuals. It is thus arguably ideal to filter not on departure from HWE (which can be caused

by either Excess Heterozygosity or Excess Homozygosity), but only on Excess Heterozygosity. Though I would not imagine that this has had any substantial impact on the analyses. But it's also the kind of filtering that is not suitable for smc++, because it's a filter that applies to variable sites but not non-variable sites.

RESPONSE: This is a good point, and one filter we didn't fully explore during the initial analysis.

In response to this comment and others concerning the smc++ analyses, we reran smc++ on a regenerated VCF without the HWE filter applied.

“Dingoes are an ancient dog breed from Asia” – I don't think dingoes should be described as a “breed”.

RESPONSE: We agree. In response to this comment, we have amended the text to read:

“The dingo, an ancient lineage from Asia...”

Reviewer #2 (Remarks to the Author):

Šlapeta and coworkers report a whole genome comparison of over 100 adult *Dirofilaria immitis* with the goal to investigate the evolutionary history of heartworms and to test the common hypotheses regarding the anthropocentric radiation of heartworms. In a principle component analysis they find four clusters, corresponding to the four continents where samples were collected. Interestingly, samples from Europe and Central America, although still discriminable, clustered together. Distinct genetic profiles could be observed for all four groups, suggesting a restricted spatial spread in the history of the species. Samples of heartworms collected from non-canids allowed for the assessment of the genetic diversity between host organisms. While such diversity can be observed, it was postulated that geography rather than host species dominated the genetic variation. Population split time analyses revealed that the divergence between populations is older than the proposed dates for dog domestication, implying that the spread of heartworms is not related to the event of dog domestication. Further analysis suggests canids like wolves or coyotes as the most likely primary hosts for heartworms in ancient times. Co-evolution of Heartworms and canids is suggested since the Oligocene. The

spread to Eurasia and South America is explained with the formation of the Bering land bridge and the Isthmus of Panama. At this time heartworms were still one population. Divergence occurred at the time of the last interglacial period. The ice sheets of the following cool period separated the ancient canid populations, and with them, potentially also the heartworm populations. Heartworm populations in Australia and Asia are related and it was postulated that the Australian Heartworm population descends from Asian populations. Introduction of heartworms to Australia is more likely to dingo hosts than to an introduction through European colonisation. A discrepancy between the estimated divergence of Asian and Australian heartworms is explained with a geographically limited sample set of Asian heartworms. Additional samples from broader geographical regions are needed. The co-clustering of European and Central American populations might be due to canid introduction during the colonisation of the Americas in the late 15th century. Split times support this hypothesis, rather than an introduction during recent migrations between Europe and Central America.

The manuscript is well written and follows a logical argumentation line. The topic is of interest to the scientific community and helps to understand the relation between different heartworm populations, which can be of interest also in the development of future heartworm preventatives and therapeutics. The data are available as well as the code used in the analysis. All programs and software suites used are described in the Methods section, allowing for the reproduction of the work.

I recommend to publish the manuscript with a minor modification (see below).

RESPONSE: We thank the reviewer for their overall positive appraisal of our work.

Lines 78-80: "This is of particular concern for domestic dogs, whereby potentially life-threatening cardiopulmonary disease is globally managed by lifelong and rigorous application of parasiticides 3." Although a treatment option for infected dogs exist, the main means is prevention using macrocyclic lactones. The therapeutic option includes the application of melarsomine, inevitably leading to thromboembolism ([https://d3ft8sckhngim2.cloudfront.net/images/AHS_Canine_Guidelinesweb22NOV2024.pdf?1732318144 \[url.au.m.mimecastprotect.com\]](https://d3ft8sckhngim2.cloudfront.net/images/AHS_Canine_Guidelinesweb22NOV2024.pdf?1732318144[url.au.m.mimecastprotect.com])). I suggest to rephrase the sentence in lines 78-80 to emphasize the prevention instead of the treatment.

Richard J. Marhöfer

RESPONSE: Thanks for the suggestion. We have rephrased this sentence to say:
“This is of particular concern in domestic dogs, where the onset of disease can be prevented through rigorous and lifelong application of parasiticides.”

Reviewer #3 (Remarks to the Author):

This is a very nice and important population genomic study which provides a new hypothesis-based framework for global population dynamics and evolutionary history of the canine heartworm *Dirofilaria immitis*. This is the largest and most thorough study of its type to date being based on whole genome sequence data from 127 individual adult worms from different carnivore hosts across four continents. The sequence data and population genomic analysis is thorough and clearly presented and the results are interesting and important.

The conclusions are very interesting suggesting a much deeper evolutionary history of *D. immitis* resulting in significant contemporary geographical sub-structuring at the continental level and supporting the hypothesis that the parasite first evolved in canids on the North American continent. Subsequent spread into Eurasia with wild canid migration across the Bering land bridge led to panmictic populations across the northern hemisphere pre-glaciation before icesheets during glaciation periods led to separation of population and genetic drift. Further hypotheses regarding subsequent spread into Australasia from Asia and a much more recent reintroduction and admixture into central America through European settlement are suggested. I am not an expert on the evolutionary analysis aspects but these hypotheses are clearly presented and seem to be well argued with appropriate caveats.

I think this paper is a good fit for *Communications Biology* and I only have a few minor critical comments:

RESPONSE: We thank the reviewer for their overall positive appraisal of our work.

Line 154 – “sample” should read “sampling”

RESPONSE: Corrected as suggested.

Lines 182-184- The data suggesting “suggesting that heartworms from canids and mustelids may be more genetically similar to each other than to those from felids” doesn’t seem that strong to me. In fact looking at extended data figure 5b the fox samples are the only ones that stand out with moderate F_{ST} s against most other hosts (except domestic dog). I would be careful not to make the case too strongly that feline *D. immitis* may be genetically different from canine *D. immitis* as per statement cited above as this would have important implications and I don’t think the data is strong enough to support it

RESPONSE: We have reworded this section to highlight the F_{ST} values of foxes, without implying that feline and canine heartworms are genetically different:

“ F_{ST} values between host pairs were generally low, with foxes showing the highest genetic differentiation from other hosts, except for the domestic dog (Supplementary Fig. 9b).”

Lines 62-64- I don’t think the overall hypothesis is well captured in the abstract –after the statement “Population structure and demographic analyses of the nuclear genome reveal distinct genetic differences between heartworms from different continents, indicating a deeper ancient origin and dispersal in canid hosts than previously recognised” the migration into Australia with dingoes and the more recent re-introduction into central America by early European colonizers is discussed, Instead, it would be helpful to have an additional sentence or two to explain the overarching hypothesis that Heartworm originated in the Americas and was panmictic with Eurasia pre-glaciation until populations were separated during glacial periods and allowed genetic separation .

RESPONSE: We have rephrased the abstract to indicate that we are testing the hypothesis that dogs were recently dispersed after domestication. We agree that we could expand more on the alternative hypothesis as the reviewer suggests; however, these are the key findings, and we want to present them as such. Further, given the limited word count of the abstract (150 words), we have decided not to elaborate as suggested.

Also, the final phrase of the final sentence of the abstract is “..which can aid future surveillance and control efforts for this important veterinary parasite”. This seems very vague and is not explained in the discussion. I suggest it is either removed or a paragraph in the discussion explaining why this work is relevant to surveillance and control efforts would be needed. eg. that molecular diagnostic tests need to account the high level of genetic differentiation globally and for the fact that genetic markers of anthelmintic resistance may not be universal applicable globally.

RESPONSE: We have removed the final phrase of the final sentence of the abstract. The final sentence now reads:

“This work sheds light on the population dynamics and deep evolutionary history of a globally widespread parasite of veterinary significance”.

Dear Editors,

RE: Decision on manuscript "Population genomics reveals an ancient origin of heartworms in canids" (COMMSBIO-25-1196A)

We would like to thank the editorial team and the reviewers for their careful consideration and overall appraisal of our manuscript. We appreciate the constructive comments and suggestions, which we have addressed in full below. We have included marked-up and clean versions of the revised manuscript to highlight the changes made.

We hope these revisions are sufficient for you to reconsider publishing our manuscript in Communications Biology.

Kind regards,

Jan Slapeta and Stephen R. Doyle
Co-corresponding authors, on behalf of all authors.

Response to reviewers

Reviewer's comments:

Reviewer #1 (Remarks to the Author):

The authors have substantially revised the part of the paper I was most sceptical about, the divergence times estimates using smc++. They performed some additional analyses and felt that these very not very reliable, and should be commended on taking the decision to leave this out, despite the large amount of work that will have gone into it. They also have revisited how they run the software for population size inferences, which now appears to be appropriate. Many of my technical concerns on this are thus now resolved. I still feel like the authors are a bit over-confident in their conclusions about the history of the heartworm, are overinterpreting the detailed shapes of Ne curves, and are perhaps overfitting their prior expectations a bit to the results. I would welcome some softening of the conclusions and acknowledgement of caveats. I offer various feedback in the below, but would overall say that I think it's a good paper that should be published, and I don't necessarily think another round of review should be necessary.

RESPONSE: Thank you.

The authors now take a different approach to estimating divergence times: they look at when effective population (N_e) size history curves start to diverge from each other. This approach was perhaps taken early in the days of PSMC, but is not so common. It's a very indirect approach.

The obvious potential limitation is that two populations sometimes might retain similar population sizes after their split – you will then not get any visible divergence in the Ne curves, despite a split having occurred. There is thus always a risk of underestimating the split time. On the other hand, if a divergence of the curves is observed, it is likely this reflects some degree of genetic separation.

RESPONSE: We appreciate the reviewer's thoughtful comment regarding the use of Ne curve divergence as a proxy for estimating population split times. We agree that this is an indirect approach and acknowledge its limitations, particularly the potential for underestimating divergence times when post-split population sizes remain similar. Our decision to explore this method was in part a response to earlier concerns raised by this reviewer regarding the challenges of interpreting split times from the data. Given the constraints of the available data and the resolution of the demographic inference methods, we aimed to provide a cautious interpretation that reflects uncertainty while still offering biological context. We have revised the manuscript to clarify that this approach is exploratory and not intended to provide precise divergence estimates. Instead, it serves to highlight potential periods of genetic separation that warrant further investigation with complementary methods and additional data.

It is in some sense a little bit problematic that, after revising their divergence time estimates quite substantially (e.g. 2-fold differences compared to first submission in some cases), the authors don't really change their conclusions. E.g. a divergence time of 29 kya and a divergence time of 10-20 kya are both taken as evidence of introduction to Australia with dingoes. I think this shows the importance of prior hypotheses in shaping conclusions. This being said, the new version of the paper is probably more likely to align with the truth.

RESPONSE: We appreciate the reviewer's observation. The revised divergence estimates (10–20 kya) actually strengthen our original hypothesis, as they align more closely with the estimated arrival of dingoes in Australia (~4 kya). Given that dingoes were the only pre-colonial canid hosts present, they remain the most plausible host for heartworm introduction. We have clarified this point in the revised text.

“Considering the variation in population demographic curves >40 kya (Figure 3a), we propose that there were substantial differences in some populations that were evolving independently before dog domestication”. There is some variation before >40 kya, but at least to me it's not clear that this is not just noise. All curves show a highly correlated pattern, first going down, and then up quite dramatically around ~50 kya - but are deviating a little bit in these periods. I would guess that degree of deviation could probably emerge as noise with smc++. Single-genome PSMC or MSMC curves are likely more robust in the distant past. As an example, there is quite some difference >40 kya between the European and Central America (CENAM) curves. The authors otherwise believe that CENAM is a very recently derived offshoot from Europe, in which case their curves should be identical >40 kya. Do the authors really believe that the Europe and CENAM curves reflect real difference in that time period? If not, I would not in general interpret

these ancient differences as reflecting independent evolution in those time periods. Personally, if I just look at Figure 3a my take-away would probably be that genuine divergences only really start at 10 kya.

RESPONSE: We appreciate the reviewer's comment and agree that Ne curve variation >40 kya should be interpreted cautiously. Our central argument is that the CENAM lineage reflects admixture between a resident American population and a recent European incursion. This could explain deviations in the curves, even in deeper time, despite the recent origin of the European component. We acknowledge that smc++ may introduce noise at these time depths, and have revised the text to clarify that this is a plausible interpretation, not a definitive conclusion.

The authors say that between Europe and CENAM "there is a striking divergence in Ne between populations around 500 years ago (Figure 3d)". Looking at Figure 3d, I would not describe this as a striking divergence. The two curves stay at very similar levels from then until the present day. There is always some minor fluctuations in these curves, but I would not interpret these results as clear evidence of genetic divergence, certainly not "striking". It is also not necessarily clear if smc++ has resolution in this recent time frame. I would focus on the genetic clustering results as the basis of the evidence for a recent European origin.

RESPONSE: We have removed the above quote of a "striking divergence in Ne between populations around 500 years ago". We now focus on the genetic clustering and shared demographic history as evidence for a recent European origin of Central American heartworms. "Finally, Europe and Central America shared considerable overlap in Ne over deep evolutionary timescales (Figure 3d)."

The authors cite an entry time for the dingo in Australia of 5-12 kya, citing dog genetic analyses – but genetic divergence times are not the same thing as entry times. 5-12 kya would be really quite early, not really aligning with the archaeological record. The earliest dog remains in Australia are from ~3.5 kya, and many would think an entry time of around 4 kya is likely. This would also align with the history of Austronesian people migrating down from Southeast Asia. The authors' new Asian-Australian heartworm divergence estimate of "10–20 kya" does still not straightforwardly align with dingo entry at 4 kya. I thus feel the conclusions on this point are a bit too strong.

RESPONSE: In response to this comment, we have updated the entry time of dingoes in Australia to "~4 kya". In our discussion, we directly acknowledge that our Asian-Australian heartworm divergence estimate of 10-20 kya does not straightforwardly align with dingo entry at 4 kya. We offer a likely explanation for this discrepancy, i.e. our Asian samples are limited to Thailand and Malaysia, which are geographically and likely genetically distinct from Indonesia, which is the likely source of dingoes entering Australia. Since dingoes did not enter Australia from Thailand or Malaysia, a discrepancy between our divergence estimate and the timing of dingo arrival in Australia is very much expected. Hence, in our discussion we suggest that

broader sampling across Asia (particularly in Island Southeast Asia) would provide further granularity in our divergence estimate. Nevertheless, we have softened the tone of our argument and have provided an alternative hypothesis for our Australian heartworms (please see below comment).

The alternative hypothesis on Australia, which to me would also seem about equally compatible with the data, is that heartworms came to Australia in the post-colonial era, from some domestic dogs brought in from Southeast Asia, China, or other parts of Asia. E.g. there are many Asian dog breeds kept as pets in Australia today, and there would have been plenty of trade and opportunity for recent introductions from different parts of Asia over the past couple of centuries. If the recent source is also from some unsampled region – for example, let's say Japan, or northern China – there would still be some amount of genetic divergence observed between the Australian heartworms and the sampled Asian heartworms in Thailand.

RESPONSE: We agree that the importation of modern dog breeds from Asia is another plausible scenario that could explain the genetic divergence between our Australian and Asian samples. In response to this comment, we have included this alternative hypothesis in our discussion.

“An alternative, and perhaps equally plausible, scenario is the post-colonial introduction of heartworms into Australia through the importation of modern Asian dog breeds. Importation of infected dogs from regions such as China or Japan could also explain the genetic divergence we currently observe between our Australian and Asian (represented only by Thailand and Malaysia) samples. Broader sampling across the Asian continent is needed to clarify whether heartworms were introduced to Australia via ancient dingoes or through the more recent importation of modern Asian dog breeds.”

The discussion on the question “If heartworms were once one population, when did they diverge, and why” is a little bit confusing to me. It mainly talks about correlated changes in Ne curves – which typically is interpreted as shared ancestry and history. It talks about glaciation potentially fragmenting heartworm populations, but this is not really seen in diverging Ne curves. The authors refer to “this fragmentation”, but fragmentation is only really clearly seen after 10,000 kya.

RESPONSE: We thank the reviewer for raising this point. We agree that the species must have originated from a single ancestral population, as speciation inherently implies a common origin. Our discussion aimed to explore when and how divergence may have occurred, using the Ne curves as one line of evidence. We acknowledge that interpreting Ne curves can be challenging, particularly given the log scale presentation, which can visually amplify recent changes while compressing deeper historical signals. This may visually obscure earlier fragmentation events that are biologically plausible but less visually prominent. The reference to “fragmentation” was intended to reflect potential population structuring events that could have occurred during glaciation periods, which may not always manifest as sharply diverging Ne curves. Instead,

correlated changes in Ne across populations may suggest shared environmental pressures or host dynamics, rather than strict demographic synchrony. We have revised the text to clarify that our interpretation is one plausible scenario, informed by the ecological history of heartworm hosts, including extinct and extant canid species, and not a definitive conclusion. We also now more clearly distinguish between inferred demographic patterns and speculative historical context.

“This time period [58–72 kya] was associated with a fall in Ne, particularly in the USA samples” – this is pretty much when the Ne curves peak to their maximal values, so slightly odd conclusion to take this as evidence for glaciation-induced declines. Overall, this section feels a bit too much like just eyeballing the Ne curves and then engaging in storytelling of a general narrative connecting the obtained results to whatever climate changes happened to occur in the time periods. I similarly think the claim of “multiple waves of dispersal with canids across the Bering Land Bridge” – I simply think this level of detail is not possible to infer just by looking at these Ne curves.

RESPONSE:

We appreciate the reviewer’s concern regarding the interpretation of the Ne curves and acknowledge that multiple plausible scenarios could explain the observed patterns. Our intention was not to overstate the certainty of glaciation-induced declines, but rather to offer one biologically plausible interpretation grounded in the parasite’s life history and ecological context. The time period [58–72 kya] does coincide with high Ne values in some populations, particularly in the USA samples, which may reflect increased transmission opportunities rather than a decline. However, we also observe shifts in Ne that could be consistent with environmental changes affecting host-parasite dynamics. We agree that the interpretation should be presented cautiously and have revised the text to clarify that this is one possible scenario among several. Regarding the suggestion that our discussion amounts to “eyeballing” the Ne curves, we respectfully note that visual inspection is a standard and necessary part of interpreting demographic reconstructions, especially when formal model-based inference is limited by the data structure. We have explicitly stated the limitations of the method and avoided overinterpreting the results. Finally, on the point about “multiple waves of dispersal with canids across the Bering Land Bridge,” we agree that the Ne curves alone cannot resolve such fine-scale dispersal events. We have revised the text to remove this specific claim and instead refer more generally to the possibility of historical host movements influencing parasite population structure.

The mitochondrial results are a little bit puzzling, as the authors say they observe very little variation in the mitochondrial genome. This would imply a very recent common ancestor of all heartworm mitochondria.

RESPONSE: As the reviewer indicated, mtDNA is indeed puzzling, as is the *Wolbachia*, but at this moment we refrain from making any speculation about the reasons or causes of it.

Reviewer #3 (Remarks to the Author):

I am satisfied with the responses to my comments and the edits to the manuscript in the revised version.

RESPONSE: Thank you.

Dear Editors,

RE: Final revisions for manuscript COMMSBIO-25-1196B

Response to reviewers

Reviewer's comments:

Reviewer #1 (Remarks to the Author):

I think the more careful caveating in response to reviewer comments has benefitted the paper. I don't think further review is needed. I just leave the authors with the below two points, where I think slight tweaks would be appropriate.

RESPONSE: We thank Reviewer 1 for their insightful comments and suggestions, which we also agree have benefited the paper.

I appreciate how the authors acknowledge that post-colonial introduction of the heartworm to Australia is also consistent with the data. However, in the abstract they still only favour the dingo hypothesis, which I thus don't think is appropriately representing the revised paper. Also, in the main text, even though they now describe the post-colonial introduction as "perhaps equally plausible", higher up they say "... dingo hosts, rather than through introduction by European colonisation".

RESPONSE: We have modified the abstract and main text to include the alternative hypothesis of a post-colonial introduction of heartworm to Australia. We have also modified Fig. 6 to include this alternative scenario.

Abstract: "Using genetic diversity and admixture analyses, we find an Asian origin for Australian heartworms, aligning with the arrival of dingoes into Australia via Asia thousands of years ago; however, we cannot exclude the alternate hypothesis that heartworms were also introduced from Asia in post-colonial times.

Main text: "Together, our findings align with an Asian origin of Australian heartworms, possibly transported with dingo hosts (Figure 3c)".

(Note: we have removed "rather than through introduction by European colonisation", and then mention the possibility of post-colonial introduction via Asian dog breeds later in the text).

In the response letter, the authors note the surprisingly low amount of diversity in the mitochondrial genome, “but at this moment we refrain from making any speculation about the reasons or causes of it.” I think it’s fine to not be able to explain everything, but from one point of view this is not ideal: one line of evidence does not align with the hypothesis of ancient diversification, so the authors do not even mention that evidence. An alternative hypothesis is that heartworm diversification is only a few hundred years deep, and the mitochondrial data reflects that, and then the deeper timescales inferred from the autosomal data are artefacts of the complicated smc++ method. I don’t necessarily think that’s the case, but I think it would be appropriate to mention with at least one sentence that the low amount of diversity in the mitochondrial genome is surprising given the proposed ancient diversification, and that the authors currently do not have an explanation for this.

RESPONSE: We agree that the low level of mitochondrial diversity is unexpected, and have included a sentence in the main text to address this.

“This contrast between the strong nuclear and weak mitochondrial population structure was particularly surprising, and the underlying biological cause remains unknown.”